
# Dual effect of anthropogenic emissions on the formation of biogenic SOA

Eetu Kari[1], Liqing Hao[1], Arttu Ylisirniö[1], Angela Buchholz[1], Ari Leskinen[1,2], Pasi Yli-Pirilä[3], Ilpo Nuutinen[3,a], Kari Kuuspalo[3,a], Jorma Jokiniemi[3], Celia L. Faiola[4,5], Siegfried Schobesberger[1], Annele Virtanen[1]

[1] Department of Applied Physics, University of Eastern Finland, Kuopio, Finland

[2] Finnish Meteorological Institute, Kuopio, Finland

[3] Department of Environmental and Biological Sciences, University of Eastern Finland, Kuopio, Finland

[4] Department of Ecology and Evolutionary Biology, University of California Irvine, Irvine, CA, United States

[5] Department of Chemistry, University of California Irvine, Irvine, CA, United States

[a] Currently working at Savonia University of Applied Sciences, Kuopio, Finland

*Correspondence to*: Annele Virtanen (annele.virtanen@uef.fi)

**Abstract.** The fraction of gasoline direct injection (GDI) vehicles comprising the total vehicle pool is projected to increase in the future. However, thorough knowledge about the influence of GDI engines on important atmospheric chemistry processes is missing—from their contribution to secondary organic aerosol (SOA) precursor emissions, SOA formation, and potential role in biogenic-anthropogenic interactions. The objectives of this study were to 1) characterize emissions from modern GDI vehicle and investigate their role in SOA formation chemistry and 2) investigate biogenic-anthropogenic interactions related to SOA formation from a mixture of GDI vehicle emissions and a model biogenic compound, α-pinene. Specifically, we studied SOA formation from modern GDI vehicle emissions during the constant load driving. In this study we show that SOA formation from GDI vehicle emissions was observed in each experiment. VOCs measured with the PTR-ToF-MS could account for 19-42% of total SOA mass generated in each experiments. This suggests there were lower volatility intermediate-VOCs (IVOCs) and semi-VOCs (SVOCs) in the GDI exhaust that likely contributed to SOA production but were not detected with the instrumentation used in this study. This study also demonstrates that two distinct mechanisms caused by anthropogenic emissions suppress α-pinene SOA mass yield. The first suppressing effect was the presence of $NO_x$. This mechanism is consistent with previous reports demonstrating suppression of biogenic SOA formation in the presence of anthropogenic emissions. Our results imply that the second suppressing effect was the presence of anthropogenic gas-phase species that suppressed biogenic SOA formation by changing the gas-phase chemistry of α-pinene. This change in oxidation pathways led to formation of α-pinene oxidation products that most likely do not have vapor pressures low enough to partition into the particle phase. Overall, the presence of gasoline vehicle exhaust caused more than 50% suppression in α-pinene SOA mass yield compared to the α-pinene SOA mass yield measured in the absence of an anthropogenic influence.





# 1 Introduction

Annual biogenic volatile organic compound (BVOC) emissions are estimated to be 825-1150 TgC $yr^{-1}$ (Fehsenfeld et al., 1992;Guenther et al., 1995;Guenther et al., 2012). In contrast, anthropogenic VOC (AVOC) emissions account for ~140 TgC $yr^{-1}$ of atmospheric VOCs (Goldstein and Galbally, 2007). Nevertheless, anthropogenic emissions are important because they

can dominate VOCs in urban areas (Warneke et al., 2007;Parrish et al., 2009;Apel et al., 2010), and are also a major source of primary particulate pollution with a global estimate of 65 Tg $yr^{-1}$ (IPCC, 2013). In addition, secondary organic aerosol (SOA) from urban sources may be the dominant source of organic aerosol at northern mid-latitudes (De Gouw and Jimenez, 2009), as it has been suggested that a large fraction of SOA from biogenic VOC indeed only forms due to the presence of anthropogenic pollution (Carlton et al., 2010;Spracklen et al., 2011).

Motor vehicles are an important anthropogenic source of not only VOCs, but also of particulate matter (PM), and of nitrogen oxides ($NO_x$). Vehicle emissions thus affect both climate and air quality, through influencing (typically enhancing) the formation of ozone ($O_3$) and of SOA (Atkinson, 2000;Monks, 2005; Gentner et al., 2017). SOA, a major component of atmospheric aerosols, is detrimental to human health but in addition to that SOA affects air quality and climate by contributing to the formation of visibility-reducing haze, and influencing the size distribution, chemical composition, and radiative and

cloud formation properties of the atmospheric particles (Kanakidou et al., 2005;Hallquist et al., 2009). As the vehicular emission standards have become stricter, the absolute emissions of the regulated pollutants, such as VOCs, carbon monoxide, $NO_x$, and PM, have substantially decreased, leading to decreases in SOA and $O_3$ formation as well (Gordon et al., 2014;May et al., 2014;Gentner et al., 2017;Saliba et al., 2017;Zhao et al., 2017;Zhao et al., 2018). However, despite the emission reductions, the newest generation of gasoline vehicles still emits substantial amounts of volatile organic and inorganic

compounds, including sulfur dioxide ($SO_2$) and $NO_x$, and once released from the tailpipe they react with the atmospheric oxidants, resulting in the formation of secondary air pollutants, such as SOA and $O_3$ (Gordon et al., 2014;Karjalainen et al., 2016;Zhao et al., 2017;Saliba et al., 2017; Zhao et al., 2018).

Gasoline vehicles can be divided into two groups based on fuel injection technologies in their engines, older technology port fuel injection (PFI) and newer technology gasoline direct injection (GDI) vehicles. GDI vehicles have better fuel efficiency

compared to PFI vehicles (Zhao et al., 1999). Due to better fuel efficiency, GDI vehicles are becoming more popular, which necessitates more detailed research about GDI vehicle emissions to better understand their impact on air quality through secondary air pollutant formation (Davis et al., 2015;Gentner et al., 2017).

Exhaust emissions from modern gasoline vehicles contribute to SOA in the atmosphere (Gentner et al., 2017;Karjalainen et al., 2016;Saliba et al., 2017;Zhao et al., 2018). The extent and details of this contribution remains a subject of ongoing research.

In addition, there are gaps in our knowledge about how new technologies specifically, such as GDI engines, influence SOA precursor emissions (Gentner et al., 2017) – crucial information for planning future vehicle emission restrictions. Only a few studies have simultaneously explored SOA formation and SOA precursors emitted from GDI vehicles. The vehicle emissions, including SOA precursors, are dependent on several factors, such as emission standard of the vehicle, fuel injection technology,



fuel used, and driving conditions (Platt et al., 2017;Peng et al., 2017;Zhao et al., 2018;Pieber et al., 2018;Du et al., 2018). Vehicles certified to stricter emission standards produce less SOA compared to vehicles certified to less strict standards due to decreased emissions of nonmethane organic gas (NMOG), including SOA precursors (Gordon et al., 2014;Zhao et al., 2017;Zhao et al., 2018). However, that concurrent reduction in SOA formation is by a smaller fraction (Gordon et al.,

2014;Zhao et al., 2017), highlighting important changes in the NMOG emission profile. Fuel injection technology may affect SOA precursor emissions from gasoline vehicles; e.g. single-ring aromatics (with low SOA mass yields) are important SOA precursors from PFI vehicles, while SVOCs and IVOCs (with higher SOA mass yields) are major SOA precursors from GDI vehicles (Du et al., 2018). However, other studies have shown no difference in SOA formation when comparing GDI and PFI vehicles in parallel (Saliba et al., 2017; Zhao et al., 2018). Aromatic content of used fuel affects greatly the emissions and SOA

formation from gasoline vehicles, as well. Recent study demonstrated that as an aromatic content of gasoline fuel increased from 29% to 37%, even 6-fold amplification of SOA production was observed (Peng et al., 2017). Moreover, driving conditions significantly affect the emissions and formed amount of SOA. For example cold-start and idling emissions are significantly higher leading to greater SOA formation compared to hot start or hot stable driving conditions (Weilenmann et al., 2009;Schifter et al., 2010;Nordin et al., 2013;May et al., 2014;Saliba et al., 2017).

Due to several factors affecting gasoline vehicle emissions, more studies under atmospherically relevant conditions with the newest generation GDI vehicles are required to better understand their effect on atmospheric chemistry and air quality. Previous studies with GDI vehicles where SOA formation and SOA precursor emissions were investigated simultaneously were conducted using different driving cycles representative for specific regions, such as Beijing, California, or Europe (Platt et al., 2017;Du et al., 2018;Pieber et al., 2018;Zhao et al., 2018). However, no studies exist where SOA formation and SOA

precursor emissions are characterized when a modern GDI vehicle is driven at constant load, although this kind of driving occurs constantly in larger roads.

In addition to anthropogenic SOA (ASOA) and O$_3$ formation, anthropogenic emissions affect atmospheric chemistry in more complicated ways by interacting with biogenic emissions, thus changing the reaction pathways of biogenic emissions from those that would occur in pristine environment. These interactions are not fully understood making it an important subject to

study. As they take place all around the world from urban to remote areas (Shilling et al., 2013;Hao et al., 2014;Kortelainen et al., 2017), these interactions need to be considered when investigating the current, future or preindustrial secondary aerosol loadings; both in interpreting atmospheric measurements and in achieving model predictions. Moreover, urban greening projects have been initiated worldwide to improve the air quality of urban areas (Escobedo et al., 2011;Calfapietra et al., 2013;Churkina et al., 2015; Ghirardo et al., 2016;Bonn et al., 2018). Due to increase of the vegetation in urban city centers,

biogenic-anthropogenic interactions will become more frequent in these areas, further highlighting the potential of these interactions to cause even more important effects on air quality and human health in the future.

Previous studies have used various approaches to explore the impact of anthropogenic emission on biogenic SOA formation and its climate-relevant characteristics. There have been field studies in environments where anthropogenic and biogenic emissions are mixing, chamber studies with relevant though typically simplified mixtures, and modelling approaches



(Goldstein et al., 2009;Carlton et al., 2010;Shilling et al., 2013;Emanuelsson et al., 2013;Hao et al., 2014). BVOCs can interact with anthropogenic emissions via several mechanisms, which are understood to varying degree and detail (Hoyle et al., 2011). For example, the increase of primary organic aerosol (POA) caused by anthropogenic emissions can enhance the partitioning of BVOC oxidation products to the particle phase if the anthropogenic POA forms a miscible phase with BVOC oxidation

products (Pankow, 1994;Odum et al., 1996;Asa-Awuku et al., 2009). In addition, anthropogenic emissions can increase the amount of acidic seed by contributing to the formation of sulfuric acid ($H_2SO_4$) and nitric acid ($HNO_3$) (Jang et al., 2002). Acidic particles catalyze heterogeneous and particle-phase reactions, such as hydration, hemiacetal and acetal formation, aldol condensation, oligomerization, and polymerization. Many of these reactions lead to low-volatility products, effectively enhancing gas-to-particle partitioning and hence SOA formation (Jang and Kamens, 2001;Jang et al., 2002;Jang et al., 2004).

As a chief precursor of atmospheric $H_2SO_4$, increased $SO_2$ levels not only increase particle acidity, but also generally enhance new particle formation and growth (Kulmala et al., 2004). Gas-phase emissions from anthropogenic sources also typically affect the gas-phase chemistry of BVOCs in the atmosphere. For example, increased concentrations of $NO_x$, CO, or aromatic VOCs can influence BVOC oxidation pathways, subsequently affecting the formation of biogenic SOA and its properties. The presence of aromatic VOCs has been observed to decrease the volatility of the formed biogenic SOA, thus affecting its

atmospheric lifetime (Emanuelsson et al., 2013). The effect of CO on α-pinene SOA formation was recently demonstrated by showing that when CO concentration is higher than 10 ppm, the dimer-to-monomer ratio of α-pinene decreases which results a suppression of α-pinene SOA formation (McFiggans et al., 2019). The impact of $NO_x$ on the reaction pathways of BVOCs and BSOA formation has been the subject of a range of previous studies. E.g., high $NO_x$ concentrations change the reaction pathway of BVOCs from $RO_2$-$RO_2$ and $RO_2$-$HO_2$ reactions toward the $RO_2$-NO reaction pathway (Presto et al., 2005;Lim and

Ziemann, 2005). The $RO_2$-NO reaction pathway leads to different functional groups in the oxidation products, compared to products observed under low $NO_x$ conditions, and may influence the degree of fragmentation occurring during the oxidative processing of BVOC. To complicate matters, these alterations in functionalization and fragmentation of BVOC oxidation products may either increase or decrease the formation of SOA, depending on the BVOC precursor. For example, the $RO_2$-NO pathway in α-pinene chemistry decreases SOA formation, but SOA formation increases when $RO_2$-NO pathway dominates

sesquiterpene chemistry (Hoffmann et al., 1997;Presto et al., 2005;Kroll et al., 2006;Ng et al., 2007a;Chan et al., 2010). The reason for this is increase still unclear, but it may be that under high $NO_x$ reaction conditions the production of large alkoxy radicals from sesquiterpenes occurs (Ng et al., 2007a). These products are less volatile than corresponding oxidation products formed under low $NO_x$ reaction conditions, which leads to more efficient partitioning of these oxidation products into the particle phase. Another reason for greater SOA formation from sesquiterpenes under high $NO_x$ reaction conditions may be that

$RO_2$+NO reaction pathway results the formation of relatively nonvolatile organic nitrates that end up to the particle phase (Ng et al., 2007a).

Several mechanisms can control anthropogenic-biogenic interactions making it a challenging subject of research. Moreover, as anthropogenic emissions are highly complex mixtures of gas- and particle-phase species, they cannot be thoroughly represented with a few model AVOCs and $NO_x$, when laboratory experiments are performed. Therefore, accurately



reproducing actual atmospheric complexities, it is crucial to study these interactions with real anthropogenic emission sources. Usage of real anthropogenic emission source enables to capture the compound complexity of anthropogenic emissions, and will lead to better understanding of the real-life interactions between biogenic and anthropogenic emissions. Moreover, more comprehensive understanding of these interactions will facilitate the estimations of the effect of these interactions on air quality

and climate. So far only one controlled laboratory study with a real anthropogenic emission source has been conducted to clarify these interactions (Kari et al., 2017). In that study, the effect of pellet boiler exhaust on α-pinene photochemistry was explored, with the conclusion that the only factor affecting α-pinene photochemistry was the high $NO_x$ conditions created by the pellet boiler emissions, thus decreasing α-pinene SOA mass yield. However, the pellet boiler emission profile differs significantly from the emission profile of GDI vehicle exhaust. For example, GDI vehicle exhaust contains significant amounts

of different VOCs, whereas pellet boiler exhaust is almost VOC free even if it is operated under nonoptimal burning conditions (Kari et al., 2017). Particle emissions are also completely different, as the pellet boiler emits much higher amounts of organic material and black carbon (BC) compared to the GDI vehicle. These large differences in emission profiles, together with the increasing use of GDI technology on our roads, encouraged us to explore if and how GDI vehicle exhaust affects α-pinene photochemistry.

This study characterizes GDI vehicle derived SOA formation from a constant load driving for the first time. For that we used GDI vehicle having the strictest emission standard certification in Europe (Euro 6) that was driven at constant load for different periods of time. Therefore, the results of this study add valuable information about the SOA formation potential of GDI vehicle exhaust, in particular when driving at constant load. In addition, our research sheds light on important interactions between anthropogenic and biogenic emissions, specifically using a gasoline vehicle and α-pinene as a model test system. Consequently,

this study's two main objectives are: 1) identify the SOA-forming VOCs emitted by a modern GDI vehicle and calculate the fraction they contribute to formed total SOA, and 2) explore how the presence of anthropogenic emissions affect the photochemistry of BVOCs, i.e. study the complex interactions taking place in regions where emissions from GDI vehicle exhaust and a model biogenic emission source are mixed.

## 2 Experimental

### 2.1 Experimental setup

Figure 1 shows the schematic of the experimental setup. The experiments were performed in a 29 m³ fluorinated ethylene propylene resin (Teflon FEP) environmental chamber located at the University of Eastern Finland in Kuopio, Finland. The chamber is a collapsible bag equipped with 40 W blacklight (BL) lamps having a spectrum centered at 340 nm (Q-Lab, 40W/UVA-340) located on two opposite sides of the chamber. The chamber is placed in a thermally insulated enclosure, and

the temperature inside the enclosure is controlled by a thermostated air conditioner (Argo AW 764 CL3). The inner walls of the temperature-controlled enclosure around the chamber are covered by reflective materials for enhancing and evenly



distributing the UV radiation. A more detailed characterization of the chamber facility is given in Leskinen et al. (Leskinen et al., 2015).

Table 1 summarizes the instrumentation used in this study. During the measurements, both gas- and particle-phases were monitored. Gas-phase VOC monitoring was done using an Ionicon proton-transfer-reaction time-of-flight mass spectrometer

(PTR-ToF-MS), acidic gas-phase species were measured using an Aerodyne acetate-chemical ionization time-of-flight-mass spectrometer (acetate-ToF-CIMS), and specific trace gas measurements included the following gases: $CO_2$, CO, $NO_x$ ($NO+NO_2$), $O_3$, and $SO_2$. Some of the most common VOCs, including 28 hydrocarbon and oxygenated hydrocarbon species (Table S1), were also measured from the raw exhaust by Fourier Transform Infrared Spectroscopy (FTIR). The chemical composition of the particle-phase was monitored with an Aerodyne soot particle-aerosol mass spectrometer (SP-AMS).

Particle mass and number concentrations and size distributions were measured using a scanning mobility particle sizer (SMPS).

### 2.2 Experimental Procedure

To achieve our research objectives described above, four kinds of photochemistry experiments were conducted (see Table 2 for more details): Pure vehicle exhaust (Pure Vehicle), α-pinene mixed with vehicle exhaust (Mixed), α-pinene with ammonium sulphate (AS) seed particles under conditions comparable to the Mixed experiments (Pure α-pin high NOx), and

α-pinene with AS seed particles under $NO_x$ free conditions (Pure α-pin NOx free).

Prior to each experiment the chamber was cleaned by evacuating the chamber and re-filling with humidified air that was produced from a zero air generator (Model 737-15, Aadco Instruments Inc., USA). The chamber was flushed with humidified air (Model FC125-240-5MP-02 Perma Pure, LLC., USA) overnight to get the chamber ready for the experiment of the next day. Also before the start of the experiment, relative humidity of the chamber was adjusted to ~50%, and the temperature was

held constant at ~20 °C with a thermostated air conditioner. As an anthropogenic emission source, a modern passenger vehicle with GDI engine (VW Golf 1.2 TSI, model 2014, Euro 6 emission standard certification) was used. Before the exhaust was fed into the chamber the vehicle was pre-heated 15 minutes prior to each experiment to warm up the engine to normal running temperature, and to exclude the cold-start emissions from the feeding. During the feeding period, the vehicle was driven at a constant load (speed 80 km/h, wheel power 23 kW) using a chassis type dynamometer (Rototest VPA-RX3 2WD). The feeding

time of the exhaust into the chamber was varied during Pure Vehicle experiments in order to study the effect of driving time on SOA precursor emissions and SOA formation (see Table 2 for feeding times). The sub-flow of the exhaust was sampled at 6 LPM from the tailpipe continuously into a dilution system while the rest of the exhaust was carried outside the building and discarded. The sampled exhaust was diluted using a two-stage dilution system consisting of a porous tube dilutor (PTD) and an ejector dilutor (ED). Zero air at room temperature was used for the dilution gas in both dilutors. . The diluted exhaust was

then introduced into the chamber. The total dilution ratios were calculated from measured $CO_2$ concentrations in the tailpipe (FTIR), and in the chamber after all dilutions (see Table 2 for total dilution ratios).

After the diluted exhaust was fed into the chamber $O_3$ was added to convert some of the NO emitted by the vehicle to $NO_2$, and then extra $NO_2$ was added into the chamber. With these additions atmospherically relevant $NO_2$-to-NO and VOC-to-$NO_x$



ratios were achieved (see Table 2) (Hoyle et al., 2011). These adjustments ensured that the important radical branching channels, such as the fate of organoperoxy radicals (RO₂) are similar to those in the atmosphere. All O₃ added into the chamber reacted immediately with NO and produced NO₂, hence no O₃ was present inside the chamber after O₃ feeding was stopped. After O₃ and NO₂ additions ~3 µl of 9-fold deuterated butanol (1-butan,$d_9$-ol, hereafter referred to as butanol-d9, Sigma-

Aldrich, 98%) and ~1µl (5 ppbv) of α-pinene (Sigma-Aldrich, ≥99%) were injected into the chamber. α-Pinene represented BVOC emissions in this study, and from the consumption of butanol-d9 the OH exposure was calculated for each experiment. Last H₂O₂ was fed into the chamber from which OH-radicals would be generated by photochemistry. After all gases and the exhaust were introduced, gases were allowed to stabilize for an additional 10 minutes, before BL-lamps were switched on to initiate photochemistry (designated the experiment start time), and the experiment was run for another 4 hours. After this

period SOA formation had stopped, so the measurements were continued with only SP-AMS for an additional hour to determine particle wall losses (see Sect. 2.3.1).

For some experiments gasoline vehicle exhaust was replaced by AS seed particles (Sigma- Aldrich, 99%) to determine the effect of GDI vehicle exhaust on the photochemistry of α-pinene. AS particles were generated by a nebulizer (Topas ATM 226, Germany) and were introduced into the chamber through a silica dryer. AS seed particle loadings were comparable with

particle loadings from GDI vehicle exhaust feeding (see Table 2). In Pure α-pin high NOx experiments all relevant conditions (i.e. VOC-to-NOₓ ratio, NO₂-to-NO ratio, and total NOₓ concentration) were comparable to Mixed experiments (see Table 2). In addition to Pure α-pin high NOx experiments, we also conducted pure α-pinene experiments under NOₓ free conditions in the presence of AS seed particles with similar OH exposure compared to other experiments to evaluate the effect of NOₓ emitted by GDI vehicle on α-pinene photochemistry. Details of the experimental procedure for the pure α-pinene experiments

can be found from the Supplemental Information (SI).

### 2.3 Data-analysis

#### 2.3.1 Correction for particle wall losses

Loss of particles to chamber walls causes significant errors when determining the amount of formed SOA from measurements if these losses are not corrected for. Therefore, the particle wall losses were determined after each experiment by monitoring

the decrease of the organic fraction of particles due to wall losses after SOA formation inside the chamber had stopped. Particle wall losses were corrected by calculating the aerosol mass loss rate constant according to Hao et al. (Hao et al., 2011).

#### 2.3.2 Predicted SOA

The GDI vehicle exhaust was comprised of a complex mixture of species, but only a subset of all these species, mostly VOCs, was detected by the PTR-ToF-MS. From the detected VOCs we identified and quantified the most important SOA precursors,

i.e. VOCs that contain at least one aromatic ring. These VOCs were considered to be the most important SOA precursors because they dominated concentration of SOA forming VOCs (> 95% in each experiment), detected by the PTR-ToF-MS.





These aromatic VOCs were used to calculate a predicted SOA mass loading for GDI vehicle exhaust. Specifically, using Eq. (1), we calculated the predicted SOA mass based on the identified SOA precursors and their literature SOA yield values for OH oxidation under high NOx conditions. Then, we used the results of Eq. (1) to evaluate how much of the observed SOA mass was predicted based on the identified SOA precursors. As per Eq. (1), the reacted concentrations of individual SOA

precursors measured by the PTR-ToF-MS were multiplied by the corresponding high $NO_x$ SOA yields of pure substances for reactions with OH-radicals found from the literature (see Table S2).

$$SOA_{predicted} = \sum_i (\Delta SOA_{precursor_i} \cdot Y_i) \tag{1}$$

where $\Delta SOA_{precursor_i}$ is the reacted concentration of $VOC_i$ measured by the PTR-ToF-MS and $Y_i$ is the high $NO_x$ SOA yield of precursor$_i$.

**2.3.3 Determination OH exposure**

OH-radical concentration and OH exposure were determined from the decay of butanol-d9 over the length of the experiment. The OH concentration was determined following the method presented by Barmet et al. (Barmet et al., 2012). OH exposure was then calculated from OH concentration by multiplying the instantaneous OH concentration by the length of the time averaging interval (TI), which was was 1 minute in this study. Eventually, OH exposure at time, t, was calculated by integrating

OH exposure from the start of the experiment until the time, t, according to Eq. (2).

$$OH\ exposure\ (t) = \sum_{i=1}^{t}(OH\ exposure_i \cdot TI) \tag{2}$$

**2.4 Mass spectrometers**

**2.4.1 PTR-ToF-MS**

VOCs were measured using the PTR-ToF-MS. Many of the trace VOCs in the atmosphere, including those emitted by the GDI

vehicle, possess higher proton affinities than water, and are consequently detectable with a PTR-ToF-MS. A detailed description of the PTR-MS has been given in several previous publications (Hansel et al., 1995;Lindinger et al., 1998;Jordan et al., 2009;Blake et al., 2009). Therefore, only key details of the PTR-ToF-MS are described in this paper. The PTR-ToF-MS consists of four main parts: 1) a hollow cathode discharge ion source, where pure ionization of water vapor generates $H_3O^+$ ions; 2) a drift tube reaction chamber, where the sampled VOCs undergo proton-transfer reactions with $H_3O^+$ ions resulting in

protonated molecules; 3) a transfer lens system that guides newly-formed ions into the mass spectrometer as a sharp beam; and 4) a time-of-flight mass spectrometer, where the protonated molecules are separated with high mass resolution based on their mass-to-charge ratios. The ions are detected by a multi-channel plate detector.

The PTR-ToF-MS instrument settings remained unchanged throughout the measurement campaign. The drift tube voltage and temperature were set to 600 V and 60 °C respectively, and the drift pressure to 2.3 mbar, which produced a reduced electric

field (E/N) of 130 Td. To minimize sampling line wall losses, the main flow from the chamber was 15 lpm through 2.5 m long





(I.D. 8 mm) unheated Teflon tubing from which the sample was continuously taken into the PTR-ToF-MS at a flow rate of 0.2 lpm through 60 °C heated PEEK (I.D. 1 mm, length 1 m) tubing.

The PTR-ToF-MS mass calibration was performed using the protonated water isotope signal (m/z 21), and signals from an internal calibrant (diiodobenzene $(C_6H_4I_2)H^+$ and its fragment ion $C_6H_5I^+$, protonated integers m/z 331 and m/z 204). PTR-

ToF-MS signals were corrected for instrumental transmission coefficients using a standard gas cylinder (AGA, Finland) containing 8 different aromatic VOCs (protonated integers ranging from m/z 79 to m/z 181). Moreover, the PTR-ToF-MS was calibrated against α-pinene with a dynamic dilution system as described in Kari et al. (Kari et al., 2018). PTR-ToF-MS data was pre-analyzed with PTR-MS Viewer version 3.2 (Ionicon Analytik, Austria), including mass calibration and peak fitting, and further analyzed using MATLAB. The concentrations of VOCs were calculated using PTR-MS Viewer software according

to the principles presented by Hansel et al. (Hansel et al., 1995). The appropriate reaction rate constants of VOCs for reactions with $H_3O^+$ ($k_{H3O+}$) were derived from Cappellin et al. (Cappellin et al., 2012). When the reaction rate constant was unknown, the collisional rate constant of $2 \cdot 10^{-9}$ cm$^3$ s$^{-1}$ was applied.

In the absence of fragmentation, sampled VOCs are measured as their protonated parent molecules ($[RH]^+$), and the high mass resolution of the instrument enables the determination of the elemental compositions of measured VOCs. However, the high

E/N value used inside the drift tube leads to fragmentation of some compounds. The degree of fragmentation depends on the E/N value applied inside the drift tube and on the structure of the compound (Erickson et al., 2014;Gueneron et al., 2015;Kari et al., 2018). For example, alcohols, terpenes, and aromatics that have larger substituents than methyl-groups undergo some degree of fragmentation. The advantage of having high E/N value inside the drift tube is reduced formation of water clusters, $H_3O^+(H_2O)$, $H_3O^+(H_2O)_2$, or $RH^+(H_2O)$, inside the PTR-ToF-MS. In this study the abundance of water-water clusters

($H_3O^+(H_2O)$ and $H_3O^+(H_2O)_2$) was less than 5% of the primary ion signal, $H_3O^+$. Hence, the proton-transfer-reactions were the major ionization reaction that took place inside the PTR-ToF-MS, and clustering of protonated molecules with water was not observed. Moreover, in this study we were able to assume that most identified vehicle exhaust SOA precursors did not go through substantial fragmentation, so the quantitation of these compounds was possible. This assumption was made based on previous literature that reports that many aromatic VOCs emitted by the gasoline vehicles do have structural forms that do not

undergo fragmentation inside the PTR-ToF-MS under the settings we operated the PTR-ToF-MS in this study (Schmitz et al., 2000;Schauer et al., 2002;Nordin et al., 2013;Platt et al., 2013;Gueneron et al., 2015). More details can be found from SI.

### 2.4.2 SP-AMS

The size-resolved chemical composition and mass concentration of aerosol were measured by SP-AMS (Canagaratna et al., 2007;Onasch et al., 2012). The detailed operational procedure of our SP-AMS was described in previous publication (Hao et

al., 2018). In brief, in the campaign, the SP-AMS was operated at 5 min saving cycles alternatively switched between electron ionization (EI) mode and SP mode. In EI mode, only the tungsten vaporizer was used to measure non-refractory chemical species. In SP mode, both the intracavity laser and tungsten vaporizers were switched on to measure laser-light-absorbing particles such as refractory black carbon (BC).




The AMS data were analyzed using the standard ToF-AMS data analysis toolkits SQUIRREL version 1.57I and PIKA version 1.16I in Igor Pro software (WaveMetrics Inc.). For the mass concentration calculations, the default relative ionization efficiency (RIE) values 1.4, 1.2, 1.3 and 1.1 were applied for organic, sulfate, chloride and nitrate, respectively. The RIE for ammonium and BC were 2.7 and 0.09, which were determined from the ionization efficiency calibration by using

monodispersed pure ammonium nitrate and black carbon (REGAL 400R pigment black, Cabot Corp.), respectively. The dataset in the EI mode was analyzed for reporting the majority of the results in the paper and also the matrices generated in that mode were used to conduct positive matrix factorization (PMF) simulations. The data in the SP mode was used to report the BC.

The PMF technique was employed to perform further analysis on the high-resolution mass spectra (Paatero and Tapper,

1994;Ulbrich et al., 2009). The PMF was evaluated with 1 to 7 factors and Fpeak from -1.0 to 1.0. After a detailed evaluation of the profiles, time series and comparison to external tracers, a four-factor solution was selected to separate the organic aerosol to four sub-groups: α-pinene_SOA_SVOOA (semi-volatile oxygenated organic aerosol) and α-pinene_SOA_LVOOA (low volatility OOA), which both originated from α-pinene SOA; HOA (hydrocarbon OA), which was dominated by the prominent $C_nH_{2n+1}^+$ and $C_nH_{2n-1}^+$ family ions and appeared as primary emission in the experiments that included the vehicle; and

mixed_SOA_LVOOA that traces to all vehicle exhaust SOA and a minor fraction from α-pinene SOA. A five-factor solution that split mix_SOA_LVOOA into two subfactors did not produce more insight and was not adopted.

### 2.4.3 Acetate-ToF-CIMS

An Aerodyne ToF-CIMS with acetate ionization scheme was used to measure acidic gas-phase species. The ToF-CIMS is described by Junninen et al. (Junninen et al., 2010) and the acetate ionization method is discussed extensively in earlier

publications (Veres et al., 2008;Bertram et al., 2011;Aljawhary et al., 2013). Briefly, the sample air into the instrument through a 2 lpm pin-hole was sampled from the main flow, which was 15 lpm through 2.5 m long (I.D. 8mm) unheated Teflon tubing. The sample air enters the Ion Molecule Reaction (IMR) region, which is held at 100 mbar pressure and where neutral gas-species are ionized and then guided to the ToF region through series of differentially pumped chambers while pressure decreases in each stage. These chambers contain two segmented quadrupoles in RF-only mode, acting as ion guides, and an

ion lens assembly.

Acetate reagent ions were generated by flowing 0.050 lpm of $N_2$ over a reservoir of acetate anhydride, diluting that flow by 2 lpm of $N_2$, and passing the diluted flow through a commercial $^{210}$Po alpha radiation emitter (P-2021, NRD) into the IMR orthogonally to the sample flow. In the IMR, an acetate anion will abstract a proton from a neutral molecule, if the neutral molecule has a higher gas phase acidity than acetic acid. In addition, acetate anions may cluster with gas-phase species, and

thus form ionized cluster adducts. Ionized molecules then enter the first segmented quadrupole region held in 2 mbar pressure, where DC voltages of individual segments are set to promote collisions of ions with surrounding neutral air molecules, causing effective breaking of clustered adducts. The ratio of the acetate reagent ion self-cluster at m/z 119 to the acetate reagent ion at m/z 59, which is often used as measure of declustering efficiency, was ~1% over the whole campaign. All measured signals



were therefore assumed to be deprotonated acids with gas-phase acidity greater than acetic acid, or possibly fragments of them due to the collisions with air molecules inside the first quadrupole.

Acetate-ToF-CIMS data post-processing was performed using the data analysis package "Tofware" (version 2.5.13, www.tofwerk.com/tofware) running in the Igor Pro environment.

## 3 Results and discussion

This study investigated emissions and SOA formation chemistry from GDI vehicles, including biogenic-anthropogenic interactions of GDI vehicle emissions related to SOA formation. In this section, first we characterize the primary emission from the GDI vehicle and demonstrate the complex nature of the exhaust. Next, we show that SOA formation from GDI vehicles was observed in each experiment, but traditional SOA precursors, i.e. identified aromatic VOCs, could not fully

explain observed SOA formation. This suggests that there were lower volatility IVOCs and SVOCs in the GDI vehicle exhaust that likely contributed to SOA production but were not detected with the instrumentation used in this study. Finally we demonstrate that the presence of GDI vehicle exhaust substantially suppressed α-pinene SOA mass yields. We will present evidences that two distinct mechanisms play a role in this suppression.

### 3.1 GDI vehicle primary emission

The GDI vehicle used in this study had the strictest emission standard certification in Europe, Euro 6. Therefore, it only emitted a small amount of primary particles and a major fraction of particle emissions was comprised of BC and only a minority of particle emissions was POA (Figure 2, panel a). In contrast to the low particle emissions, GDI vehicle exhaust comprised a significant amount of $NO_x$ (see Table 2) and a complex mixture of VOCs. Initial $NO_x$ in the chamber was mainly NO, but some was converted to $NO_2$ to obtain an atmospherically relevant $NO$-to-$NO_2$ ratio (see Sect. 2.2). Figure 2 (panel b) shows

the average gas-phase mass spectrum measured from the chamber after the feeding period by the PTR-ToF-MS. The exact mass of each protonated ion was used to assign elemental composition to the major peaks. SOA precursors were identified (Table S2) via a process that combined information from elemental composition with previously-identified compounds from gasoline vehicle emissions (Schauer et al., 2002;Erickson et al., 2014;Gueneron et al., 2015;Pieber et al., 2018), from which we proposed likely molecular structures. As Figure 2 shows, in addition to SOA precursors, GDI vehicle exhaust comprised a

great number of small VOCs that were mainly hydrocarbons or oxygenated hydrocarbons, such as $C_3H_5^+$ (m/z 41), $C_2H_2O$ and $C_3H_7^+$ (m/z 43), acetaldehyde (m/z 45), formic acid (m/z 47), acrolein and $C_4H_9^+$ (m/z 57), acetone (m/z 59), acetic acid (m/z 61), methacrolein (m/z 71), and methyl ethyl ketone (m/z 73)– consistent with compounds and fragments identified from gasoline vehicle exhaust previously (Schauer et al., 2002;Platt et al., 2013;May et al., 2014;Gueneron et al., 2015;Pieber et al., 2018). After the exhaust had been fed into the chamber, the conditions inside the chamber were, regarding the gas-phase,

reflecting ambient observations at urban sites (e.g. Barcelona or Rome (Seco et al., 2013;Fanizza et al., 2014)).





### 3.2 Photochemistry of GDI

Figure 3 shows a time-series of the gas and particle-phase compounds during a typical Pure Vehicle photochemistry experiment. Photochemistry started at time 0. Different aromatic VOCs reacted with OH-radicals at different reaction rates (Figure 3, panel a). For example, the decay of toluene is substantially slower compared to the decay of xylene or

trimethylbenzene (TMB). SOA precursors reacted with OH-radicals to form gas-phase oxidation products, such as methyl ethyl ketone (MEK), $C_4H_2O_3$, and $C_4H_4O_3$, and particle-phase organics shown in the figure as OA (green diamonds in Figure 3). The mass of SOA generated was substantially higher than the POA mass emitted by the GDI vehicle: SOA-to-POA ratios varied from 14 to 35 in Pure Vehicle experiments. These ratios are in agreement with previous studies where SOA-to-POA ratios were higher than 10 in experiments using older GDI vehicle exhaust (Karjalainen et al., 2016;Pieber et al., 2018). The

panel b of Figure 3 shows that at the beginning of the experiment both $NO_x$ species were present in the chamber from which NO was converted rapidly to $NO_2$ when the photochemistry was started. Due to the presence of significant amounts of $NO_x$, more than 20 ppb in each experiment (see Table 2), we classified our GDI vehicle experiments as "high $NO_x$" experiments. The presence of NO initiates an additional reaction pathway of $RO_2$ + NO that leads to formation of VOC reaction products that likely have too high vapor pressures to end up in the particle phase (Presto et al., 2005;Ng et al., 2007b). This additional

reaction pathway causes the delay in the start of SOA formation as Figure 3 shows– SOA formation did not start until the most of NO had reacted. The panel b of Figure 3 also demonstrates that after the lights were turned on, the photochemistry of $NO_x$ and VOCs produced 200 ppb of $O_3$.

We identified 20 aromatic species as ASOA precursors in the PTR-ToF-MS data, and with Eq. (1) we calculated the predicted SOA mass generated by the reactions of these species with OH radicals. Predicted SOA mass based on these precursors could

explain 19-42% of the measured SOA (Figure 4). In Figure 4, the ASOA precursors were divided into six different groups based on elemental composition and number of carbon atoms: benzene, C8 aromatics, C9 aromatics, toluene, oxygenated aromatics, and other aromatics. Apparent from Fig. 4 (stacked bars) is the variability between individual experiments, both in the different groups of SOA precursors and in the amount of SOA formed by their oxidative processing. The variations may simply be related to experiment-to-experiment variability in GDI vehicle emissions. In any case, however, Figure 4

demonstrates that we observed much more SOA (circles) than predicted based on the identified SOA precursors, implying that we were missing a substantial fraction of SOA precursors in our data-analysis. Error bars of the predicted SOA (stacked bars) in Figure 4 show the lower and upper bound estimates. These were calculated by taking the lowest and the highest reported SOA yield value of each aromatic VOC from the literature, reported under similar conditions compared to our study (see applied yields from Table S2). Hence, this gap between predicted and observed SOA cannot be explained by SOA mass yields

applied to calculate predicted SOA values. Previously it has been shown that IVOCs alone create about 50% of the formed SOA from the gasoline vehicle exhaust (Zhao et al., 2016). The same observation was previously reported in a more recent publication, where comprehensive organic emission profiles for relatively new gasoline vehicles were constructed based on previous studies (Lu et al., 2018). The conclusion of that study was that  IVOCs and SVOCs contribute approximately 50% to



the formed SOA from gasoline vehicle exhaust, even if VOCs clearly dominate the emission profile (Lu et al., 2018). Based on the results of these previous studies, we can assume that also in our study GDI vehicle emitted IVOCs and SVOCs, and these emissions made a substantial contribution to formed SOA. This result is consistent with other studies that have explored modern gasoline vehicle emissions during driving cycles, which do not represent constant load driving that was used in this

study (Gordon et al., 2014;Zhao et al., 2016;Zhao et al., 2017). These IVOCs and SVOCs have high SOA mass yields. For example, for linear, cyclic, and branched 12-carbon alkanes SOA mass yields can vary from 11% up to 160% under high $NO_x$ conditions (Loza et al., 2014). However, these SOA precursors were not detected by the PTR-ToF-MS, most likely due to sampling line and instrumental losses (Pagonis et al., 2017). Hence, the missed detection of these compounds in our study could explain the significant gap between predicted and measured SOA shown in Figure 4.

The gap between measured and predicted SOA increases with exhaust feeding time; and interestingly it does so much more strongly than the increase in predicted SOA (stacked bars). The vehicle heating procedure prior to the exhaust feeding period was the same in each experiment and the vehicle was driven at a constant load for all experiments, so the longer feeding time corresponds to longer driving time. Longer driving time decreased the predicted/measured SOA ratio indicating there was a larger fraction of unaccounted SOA precursor in the chamber. As argued above, we attribute the missing SOA precursor to

increased IVOCs and SVOCs that were not detected with the PTR-ToF-MS. This is an important and interesting finding, and to our knowledge, this has not been reported earlier. Previous studies with relatively new gasoline vehicles have reported that IVOC and SVOC species together would contribute as much to formed SOA as VOC species (Zhao et al., 2016;Zhao et al., 2017;Lu et al., 2018). Our results imply that the contribution of IVOCs and SVOCs to formed SOA is driving time dependent, at least when the modern gasoline vehicle is driven at constant load.

Zhao et al. found that IVOC emissions as a fraction of NMOG emissions are enriched during the hot operation compared to cold-start operation (Zhao et al., 2016). This may be due to higher combustion efficiency of more volatile fuel components, when the catalyst has reached its optimal temperature. Recently, Pereira et al. showed that IVOCs are removed less efficiently than VOCs by catalytic converters in diesel vehicles (Pereira et al., 2018). Similarly, we hypothesize that the catalysts in gasoline vehicles (like the vehicle used in this study) remove IVOCs and SVOCs less efficiently than lower-molecular-weight

organics. This would essentially enrich the vehicle exhaust with IVOCs and SVOCs, which have oxidation products with higher SOA mass yields. Moreover, the condensation of IVOCs and SVOCs to tubings of the vehicle that are then released by outgassing when the vehicle is driven for longer time could explain the increasing fraction of IVOCs and SVOCs from total emissions as a function of driving time. However, this must be studied more in future. For example, future studies must characterize the emitted SVOCs and IVOCs to see if their chemical composition changes throughout the driving period or if

the same compounds are emitted by the vehicle with increasing quantity that would support the hypothesis about the condensation of IVOCs and SVOCs to tubing of the vehicle and to transfer lines (Pagonis et al., 2017;Deming, 2019). In our case the feeding time dependence of IVOC/ SVOC losses (or outgassing) may be affected by time dependent changes in tubing and transfer line temperature caused by the hot exhaust. Any induced errors would lead to an underestimation of emissions-



induced SOA yields, which means that chamber studies in general underestimate SOA yields, and formed SOA mass, from a vehicle exhaust due to line losses of these high-yield SOA precursors.

The results of this study offer new useful information from GDI vehicle emissions during constant load driving that have not been considered in previous studies. To our knowledge, this is the first time that the chemistry of modern gasoline vehicle emissions were explored during constant load driving and with a pre-heated engine. This is in contrast to studies using cold-start emissions combined with breaking and acceleration periods—conditions which generally produce higher emissions from the vehicle, but do not necessarily represent realistic emissions for most vehicles on the road. This difference in study design makes it impractical to compare the absolute mass of SOA formed between studies. However, predicted SOA offers a more suitable value for comparison purposes, i.e. to evaluate if our results are in agreement with previously published results of GDI and PFI vehicle emission studies under controlled conditions. In those previous studies, emitted VOCs identified as SOA precursors, have contributed from less than 20% up to 100% to the measured SOA (Platt et al., 2013; Nordin et al., 2013; Gordon et al., 2014;Liu et al., 2015;Peng et al., 2017;Du et al., 2018;Pieber et al., 2018). Our results (19%-42%) fall within that reported range even if the driving conditions and emission standard certification of the vehicles were different between the various studies. However, some studies have reported a much smaller "missing source" of SOA precursors. For example, 60% of observed SOA mass generated from idling phase emissions could be accounted for by the reactions of $C_6$-$C_9$-aromatics (Nordin et al., 2013), and even 90% of the SOA formed during the idling period was explained by the reactions of benzene, toluene, $C_8$-$C_{10}$-aromatics and naphthalene (Liu et al., 2015). During those studies, the vehicle was kept outside the laboratory in ambient temperature conditions. Furthermore, another study using gasoline vehicles manufactured prior to 1995 has shown that traditional SOA precursors, such as single ring aromatics and mid-weight VOCs ($C_9$-$C_{12}$), could fully explain the SOA formed from oxidation of emissions during cold-start (Gordon et al., 2014). That same study demonstrated that unspeciated organics, i.e. the organics that were not identified but detected as a part of total NMOG, were responsible for the majority of the SOA formed from newer vehicles during the same driving cycle (Gordon et al., 2014). Therefore, when comparing vehicle derived SOA formation and SOA precursor emissions between different experiments, it is crucial to know the experimental conditions, including the type of emissions (idling, cold start, hot start etc.), and the specifications of a vehicle. These factors most likely explain the most of the discrepancy observed between the results of different studies.

Predicted SOA is greatly affected by SOA yield values (low or high $NO_x$ regime) applied for the calculation. Different research groups have justified the use of certain SOA yields differently. Some groups have just examined NO levels, and when NO concentration has reached nearly 0 ppb concentration after photo-oxidation period has continued for a while (exact time depends on the study and authors how they define that), they have argued that their experiments have mainly took place under low $NO_x$ conditions ( Platt et al., 2013;Liu et al., 2015;Pieber et al., 2018). In contrast, other groups, including us, have used the VOC-to-$NO_x$ ratio to define conditions (Zhao et al., 2016;Peng et al., 2017;Du et al., 2018). The calculation of predicted SOA is highly sensitive to which SOA yield values are used for different SOA precursors because SOA yields under high- and low-$NO_x$ are significantly different for several SOA precursors, including aromatic VOCs (Ng et al., 2007b).



The largest uncertainty in predicting SOA mass is caused by the missing yield values of some SOA precursors identified from the exhaust. In this study, predicted SOA was calculated using literature SOA yields for all aromatic VOCs using studies with the most similar aerosol mass concentration as observed in this study because SOA production is influenced by the gas-particle partitioning (Odum et al., 1996). Unfortunately, SOA yields for all identified SOA precursors were not available, and in those

cases, SOA yields of other compounds with similar structures were used as a proxy. In this study we made every effort to include the full range of possible SOA mass yield values, which provides increased confidence that the ranges reported here actually represent a lower and upper bound for the predicted SOA. In spite of these uncertainties in predicting SOA mass, our calculations provide useful information about the missing fraction of SOA precursors and their relative contribution to SOA formation in different experiments during this study.

In summary, the following reasons can explain the significant differences between predicted and measured SOA. First, as explained above, uncertainties in the calculations to predict SOA may cause some of the difference. However, two more significant issues are: 1) modern GDI vehicles emit SOA-forming IVOCs and SVOCs that have high SOA yields, and cannot be detected by the PTR-ToF-MS; and 2) the photo-oxidation of VOCs in a more complex mixture may also result more complex reaction pathways and product distributions, and hence an altered SOA yield compared to pure single precursor

experiments (Song et al., 2007). To better understand SOA formation from vehicle exhaust, instrumentation that can measure IVOCs and SVOCs is crucial to be included in future experimental setups. Identifying those compounds will also help us understand why their fractions of total emissions increase with increasing constant load driving time.

**3.3 Separation of anthropogenic and biogenic SOA in Mixed experiments**

Vehicle exhaust derived SOA formation complicates the estimation of α-pinene derived SOA formation during Mixed

experiments. Therefore, to estimate if the presence of GDI vehicle exhaust affects the photochemistry and SOA formation of α-pinene, we must separate ASOA from biogenic SOA (BSOA) in Mixed experiments. For that we applied three independent methods, which were used to estimate ASOA formation in Mixed experiments from GDI vehicle exhaust. These three methods are described next.

In the first method we applied PMF analysis to AMS data to quantify the fraction of formed SOA originating from

anthropogenic precursors. As shown in Figure S1 and explained above in Sect. 2.4.2, PMF analysis conducted for all experiments produced a four-factor solution. Previously it was shown that gasoline vehicle emitted IVOCs and POA are correlated (Zhao et al., 2016). Therefore, based on the observed correlation between IVOCs and POA, we estimated the amount of formed ASOA from the linear fit of the HOA factor as a function of the mix_SOA_LVOOA factor shown in Figure S2. We justify this approach by the strong observed contribution of IVOCs to formed ASOA, and the fraction of ASOA (out of total

SOA) should be strongly reflected in the mix_SOA_LVOOA factor. Hence, by using PMF results of the four Pure Vehicle experiments to create a linear fit, we were able to use the equation of that fit ($r^2$=0.97) to estimate the amount of formed ASOA during Mixed experiments.



In the second method we used a reacted concentration of a single ASOA precursor having an elemental composition of $C_9H_{10}$ to assess the formed ASOA in Mixed experiments. In this method we assumed that the reactions between $C_9H_{10}$ and OH-radicals contributed equal fractions to the formed SOA in each experiment. By fitting the formed SOA as a function of the reacted amount of $C_9H_{10}$ in Pure Vehicle experiments, we estimated from the linear fit equation ($r^2$=0.98) the formed ASOA

in Mixed experiments (Figure S3). We tested different ASOA precursors for the linear fit but $C_9H_{10}$ gave the best correlation against the formed SOA, so it was chosen.

In the third method, we first calculated effective SOA yield of the vehicle exhaust using Eq. (3):

$$Y_{effective} = \frac{\Delta M_0}{\Sigma_i \Delta SOA\_precursor_i} \tag{3}$$

where $\Delta M_0$ is the maximum mass concentration of SOA formed (maximum OA minus primary OA mass), $\Sigma_i \Delta SOA\_precursor_i$

is the sum of the reacted concentration of all identified SOA precursors shown in Table S2.

With the effective SOA yield, we estimated the formed ASOA in the Mixed experiments based on the total concentration of reacted ASOA precursors measured with the PTR-ToF-MS. For this method we assumed that the distribution of isomers of each PTR-ToF-MS detected elemental composition in the exhaust did not change between the experiments, because different isomers undergo different fragmentation inside the PTR-ToF-MS, which interferes with their quantitation, but they would have

different SOA mass yields. Based on previous studies this was a reasonable assumption (Schauer et al., 2002;Gueneron et al., 2015;Schmitz et al., 2000). The average effective SOA yield of Pure Vehicle1 and Pure Vehicle2 experiments was used in this calculation as effective ASOA yield value for Mixed experiments. The average effective SOA yield was applied because the feeding times of the vehicle exhaust in Mixed experiments were between the feeding times of Pure Vehicle1 and Pure Vehicle2 experiments (see Table 2).

From all these three methods, independent from each other, we obtained estimates of the formed ASOA in the Mixed experiments that were relatively close to each other (Table 3). This agreement gives confidence that the average ASOA determined from these three methods represents well the formed ASOA in Mixed experiments. As Table 3 shows, in the Mixed experiments α-pinene derived SOA dominated the total SOA mass but the fraction of the vehicle exhaust derived SOA was significant and needed consideration in estimating α-pinene derived SOA.

**3.4 Dual effect of anthropogenic emissions on biogenic SOA formation**

In this study our second objective was to explore the effect of anthropogenic emissions on biogenic SOA formation. To study this, we used gasoline vehicle exhaust and α-pinene as a model test system, and we compared α-pinene SOA mass yield between Mixed and Pure α-pin experiments under high $NO_x$ and $NO_x$ free conditions. α-Pinene SOA mass yield was calculated by Eq. (4).

$$Y_{a-pinene} = \frac{\Delta M_{0\,a-pinene}}{\Delta[a-pinene]} \tag{4}$$





where $\Delta M_{0\ \alpha\text{-pinene}}$ is the maximum wall loss corrected mass concentration of SOA formed due to photo-oxidation of α-pinene, and $\Delta[\alpha\text{-pinene}]$ is the mass concentration of α-pinene reacted. In mixed experiments $\Delta M_{0\ \alpha\text{-pinene}}$ was calculated by subtracting ASOA from total SOA, as described above.

Figure 5 shows that in the presence of gasoline vehicle exhaust α-pinene SOA mass yield was lower compared to

comparable Pure α-pin experiments under both conditions (high $NO_x$ and $NO_x$ free). Clearly, Figure 5 demonstrates the dual effect of anthropogenic emissions on biogenic SOA formation. Figure 5 shows that the gasoline vehicle exhaust significantly suppressed α-pinene SOA mass yield when compared to α-pinene mass yields in the absence of an anthropogenic influence (red circles compared to black triangles). The maximum α-pinene SOA mass yields observed for Mixed experiments ranged from 18.7±2.2% to 26.8±1.4%, whereas for Pure α-pin NOx free experiments mass yields ranged from 42.2±1.9% to

42.3±1.5%. Therefore, in the presence of the gasoline vehicle exhaust 37%-56% suppression in α-pinene SOA mass yields were observed. The results suggest that $NO_x$ emitted by the gasoline vehicle was the most important factor in α-pinene SOA mass yield suppression. This is demonstrated by Pure α-pin experiments under high $NO_x$ and $NO_x$ free conditions (blue squares compared to black triangles). From now on we call this the "$NO_x$ effect". The maximum α-pinene SOA mass yields observed for Pure α-pin high NOx experiments ranged from 23.8±1.9% to 29.5±1.5%. Moreover, Figure 5 shows that the α-

pinene SOA mass yield was lower yet in the Mixed experiments compared to the Pure α-pin High NOx experiments, which featured comparable $NO_x$ levels, in particular when the initial surface area of particles was similar between the experiments. We interpreted the difference between the two results to be an "anthropogenic VOC effect" because $NO_x$ concentration levels were similar in those experiments (see Table 2) and thus "$NO_x$ effect" on the mass effect could be ruled out. Suppression of α-pinene SOA mass yields attributed to the "anthropogenic VOC effect" varied from 9.2% to 21.4%. It

should be noted that in the Mixed experiments surface area of particles was higher than in comparable Pure α-pin High NOx experiment. Hence, if surface area of particles would have been equal in both experiments the SOA mass yield difference would have been even higher.

Figure 6 shows the evolution of α-pinene SOA mass yields as a function of OH exposure. It shows that in every experiment where $NO_x$ was present, higher OH exposure was required before SOA formation started. The similar observation that higher

OH exposure under high $NO_x$ conditions is required before SOA formation starts during α-pinene photo-oxidation was also reported in previous study (Eddingsaas et al., 2012). We attribute this behavior mainly to the presence of NO that inhibited the start of SOA formation until most of NO had reacted away, as was explained above (Sect. 3.2). This effect of $NO_x$ on the formation of biogenic SOA, in particular α-pinene SOA, has been reported repeatedly in previous studies, also with real anthropogenic emission sources (Presto et al., 2005;Ng et al., 2007a; Eddingsaas et al., 2012;Kari et al., 2017). Previous study

showed that the "$NO_x$ effect" clearly suppressed α-pinene SOA formation in the presence of pellet boiler exhaust, and that this effect was the only suppressing factor in that system (Kari et al., 2017). Interestingly, Figure 6 also shows that the α-pinene SOA formation was even more delayed in the Mixed experiments compared to the Pure α-pin high NOx experiments. This additional delay in SOA formation has not been observed previously, and it offers further evidence that, in addition to the "$NO_x$ effect", also some additional mechanism affects the α-pinene SOA formation in the presence of the vehicle exhaust. This

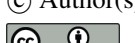



mechanism both eventually results the suppression of α-pinene SOA mass yield (Fig. 5), and delays α-pinene SOA formation in our chamber (Fig. 6). It should be noted that based on the conclusion made by Kari et al. about the interactions between α-pinene and pellet boiler exhaust (Kari et al., 2017), we can conclude that the additional mechanism is really caused by VOC and not primary particles, hence we call it "anthropogenic VOC effect". Specifically, Kari et al. showed that the observed α-

pinene SOA mass yields were similar regardless of the primary particle material (pellet boiler emitted particles or AS seed particles).

Elevated $NO_x$ concentrations reduce $RO_2+RO_2$ and $RO_2+HO_2$ reactions while initiating $RO_2+NO$ reactions, as was explained above. This shift in reaction pathways of α-pinene produces organonitrate oxidation products with higher vapor pressures compared to oxidation products generated under the $NO_x$ free conditions (Ng et al., 2007a). Despite their higher vapor

pressures, some organonitrates will partition into the particle phase. Based on our estimation on the neutralization of SOA formation in Mixed and Pure α-pin experiments, the majority of particulate nitrate was present in the chemical form of ammonium nitrate. However, we also observed typical signs of the appearance of organonitrates in the particle phase (Farmer et al., 2010). We observed the ratios of $NO_2^+$-to- $NO^+$, which is one of variables potentially indicating the chemical forms of nitrate, to monotonically increase from 1.5 to 2.0 after BL lamps were switched on in the Mixed experiments. The ratio of

$NO_2^+$-to- $NO^+$ for $NH_4NO_3$ is 1.46±0.02 (Farmer et al., 2010). The ratios of $NO_2^+$-to- $NO^+$ measured here were about 3 in Pure α-pin high-NOx experiments, vs. 2 in Pure α-pin NOx free experiments. Hence, these results suggest organonitrate formation after photooxidation reactions took place inside the chamber, providing observational evidence of the elevated $NO_x$ indeed altering reaction pathways of α-pinene during photooxidation, likely decreasing α-pinene SOA mass yields.

Figure S4 demonstrates that in Mixed experiments we observed the formation of oxidation products that were absent in Pure

vehicle and Pure α-pin high NOx experiments. As these oxidation products were only present in Mixed experiments, α-pinene derived intermediate reaction products must have reacted with vehicle originated gas-phase species that has thus changed the reaction pathways of α-pinene. The formation of these oxidation products most likely leads to an α-pinene product distribution with overall higher vapor pressures compared to the oxidation product distribution formed in the absence of gasoline vehicle exhaust, yet presence of $NO_x$ (Pure α-pin high NOx experiments). That way, the formation of new oxidation products could

explain the smaller SOA mass yields of α-pinene observed in Mixed compared to Pure α-pin high NOx experiments. The identification of the exact elemental compositions of the detected oxidation products is difficult, because we only have information about the m/z of the detected ions. Although acetate-ToF-CIMS has high mass resolution (> 4000), still more than one possible elemental composition can fit into the same m/z within the precision that can be reached with this instrument. Therefore, only acetate-ToF-CIMS detected (m/z)s of these oxidation products are given in Figure S4.

Our results imply that gas-phase species emitted by gasoline vehicle may play the most important part in the "anthropogenic VOC effect" by changing the reaction pathways of α-pinene that results the formation of new α-pinene oxidation products. However, with the instrumentation included in this study we cannot rule out other possible mechanisms that may have decreased SOA mass yield of α-pinene in the presence of vehicle exhaust. An additional explanation for "anthropogenic VOC effect" that decreases SOA mass yield of α-pinene could be the inhibition of the formation of SOA-forming highly oxygenated





molecule (HOM)-dimers from α-pinene photo-oxidation caused by the vehicle exhaust. It was recently shown that α-pinene SOA mass yield was suppressed due to inhibition of α-pinene HOM-dimer formation in the presence of isoprene, CO, and methane (McFiggans et al., 2019). Recently published article demonstrated that the presence of peroxy radicals from additional compounds resulted in termination reactions of α-pinene oxidation products and thereby effectively prevented the formation

of α-pinene HOM-dimers that suppressed α-pinene SOA mass yield (McFiggans et al., 2019). It should be noted that, the effect of CO on α-pinene SOA mass yield reduction in our study has been significantly smaller compared to the study of McFiggans et al. because we had substantially less CO introduced into the chamber (< 1 ppm compared to ≥ 10 ppm) (McFiggans et al., 2019). Moreover, the reacted concentration of α-pinene was twice as high as in the study of McFiggans et al. compared to our study (~10 ppb compared to ~5 ppb), and the reacted concentration of isoprene (~10 ppb) in their study was ~5-fold higher

compared to the sum of reacted concentrations of vehicle emitted VOCs (~2 ppb), as detected by the PTR-ToF-MS, in our study. However, the differences in VOC concentrations between the studies were clearly smaller than the differences in CO concentrations. Therefore, if α-pinene HOM-dimer formation was reduced in our study, and the cause of the "anthropogenic VOC effect", we would assume that the main contributor to that was the vehicle emitted VOC mixture, i.e. that that mixture acted in an analogous way to isoprene in the previous study (McFiggans et al., 2019). This potential scenario is in line with

their conclusion that their results can be extended from the mixtures they used to other reactive atmospheric mixtures of vapors (McFiggans et al., 2019). Unfortunately, during our measurement campaign we were not able to measure low-volatility gas-phase species. To verify if the vehicle exhaust inhibits HOM-dimer formation from α-pinene photo-oxidation, more studies with instruments that can measure HOM-dimers are required.

We acknowledge that our estimations of formed ASOA in Mixed experiments causes uncertainty for the α-pinene SOA mass

yield calculation. To remove this uncertainty from our analysis, we did a linear combination analysis that provides additional piece of evidence that some other mechanism in addition to $NO_x$ plays a role in the suppression of α-pinene SOA formation in the presence of gasoline vehicle exhaust. Results and discussion about the linear combination analysis can be found from SI and Figure S5.

It should be noted that it has been shown that SOA yields can be underestimated due to wall losses of SOA forming vapors

during chamber experiments (Kokkola et al., 2014;Zhang et al., 2014;Zhang et al., 2015;Yeh and Ziemann, 2015; Platt et al., 2017). For this reason the SOA mass yields and the amounts of formed SOA presented in this paper may be lower limits. Figure 5 provides insight into the effect of vapor wall losses on SOA mass yield, as the yield increases as a function of initial surface area of particles. I.e., the observed yields are dependent on the condensation surface area of the seed particles present inside the chamber during the experiment. Obviously, the seed surface area available for the vapors to condense on affects the

yield due to competition for vapors between chamber wall surface and particle surface. However, this does not change the conclusion of this work that two distinct mechanisms caused by anthropogenic emissions suppress biogenic SOA formation, because all experiments were conducted using the same chamber and similar initial surface area of particle loadings.



## 4 Conclusions

This study had two main objectives. First we studied SOA formation chemistry from emissions of modern GDI vehicle during constant load representing typical freeway-driving conditions. Second, we explored the complex chemical interactions between anthropogenic and biogenic emissions by using GDI vehicle exhaust and α-pinene as a model test system. We showed that

anthropogenic SOA precursors identified from GDI vehicle exhaust by PTR-ToF-MS contributed 19-42% to the measured SOA. This suggests there is a missing source of vapors in the emissions that are contributing to SOA production. This missing source of SOA precursor vapors is larger when driving time was longer, which can be explained with an increased fraction of IVOCs and SVOCs being transported to the environmental chamber with increased driving time. To our knowledge, an increasing fraction of IVOCs and SVOCs from total emissions as a function of driving time has not been reported before. This

finding illustrates one of the unknown aspects from modern vehicle emissions that needs to be clarified in order to better understand the effect of vehicle emissions on climate and air quality. These results agree with previous studies that clearly showed that the current vehicle evaluation system, which mainly measures the emissions of PM, total hydrocarbons (THC), $NO_x$, and CO, will not be sufficient in the future, because the vehicles also produce IVOC and SVOC emissions that play a great role in vehicle derived SOA formation. Consequently, despite stricter emission regulations, the substantial amounts of

gasoline vehicle derived SOA will be formed in several regions in the future, unless the vehicle evaluation systems and emission regulations are updated.

This study also demonstrated two mechanisms by which anthropogenic emissions suppressed biogenic SOA formation– the well-established "$NO_x$ effect" and a newly-proposed "anthropogenic VOC effect". Our results suggest that anthropogenic emissions can substantially decrease biogenic SOA formation in urban regions compared to biogenic SOA formation in pristine

environments. Compared to pristine environments (here represented by α-pin NOx free experiments) α-pinene SOA mass yields were 37%-56% lower due to an anthropogenic influence. Moreover, compared to the environment where the only anthropogenic influence would be the presence of $NO_x$ (represented by α-pin high NOx experiments), α-pinene SOA mass yields were 9.2%-21.4% lower due to the "anthropogenic VOC effect" caused by the presence of the vehicle exhaust. Further, our results imply that gas-phase species emitted by the gasoline vehicle may play an important part in the "anthropogenic VOC

effect". In Mixed experiments, we observed the formation of new oxidation products of α-pinene that most likely have higher vapor pressures compared to α-pinene oxidation products in the absence of anthropogenic emissions. These higher vapor pressure oxidation products would undergo less efficient gas-particle partitioning, and would ultimately decrease SOA mass yields from α-pinene oxidation. However, with the instrumentation included in this study we cannot rule out other possible mechanisms behind the "anthropogenic VOC effect", such as inhibition of α-pinene HOM-dimer formation, that may have

also contributed to reduced α-pinene SOA mass yields in the presence of vehicle exhaust. In future studies, it should be clarified if the "anthropogenic VOC effect" suppresses SOA formation of other types of biogenic VOCs, as well, in particular other monoterpenes, sesquiterpenes and isoprene. That clarification would help us draw a more general picture about the broader




influence of the "anthropogenic VOC effect" on SOA formation in regions where biogenic and anthropogenic emissions are mixed.

## Author's contribution

EK and AV designed the study; EK, LH, AY, AB, AL, PY, IN, KK, and JJ conducted the experiments; EK, LH, AY, AB, IN,

CLF, SS, and AV participated in data analysis and/ or interpretation; EK wrote the manuscript; LH, CLF, SS, and AV edited the manuscript.

## Acknowledgements

This work was financially supported by the European Research Council (ERC Starting Grant 335478) and The Academy of

Finland Center of Excellence programme (grant No 307331). This work has received funding from the European Union's Horizon 2020 research and innovation programme through the EUROCHAMP-2020 Infrastructure Activity (grant No 730997). The work of E. Kari was financially supported by University of Eastern Finland Doctoral Program in Environmental Physics, Health and Biology.

Otso Peräkylä and Mikael Ehn from INAR (Institute for atmospheric and Earth system research) are thanked for the help they

provided with acetate-ToF-CIMS operation during the measurement campaign.

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

**Figures and Tables**

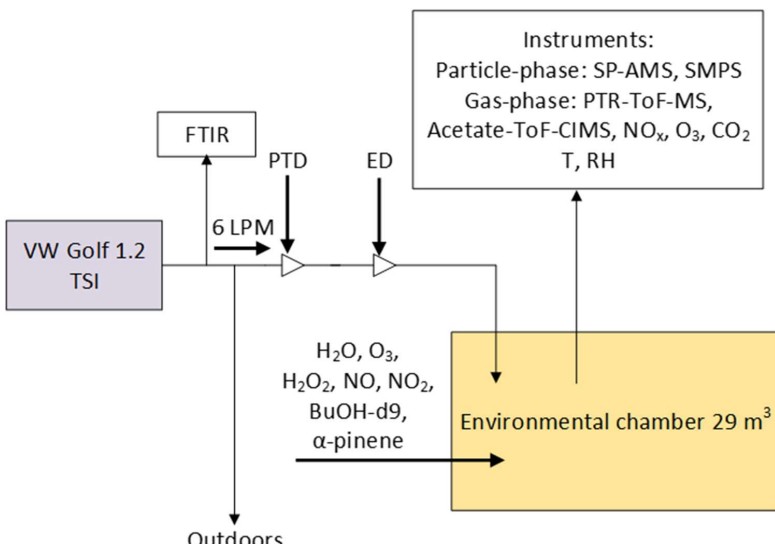

**Figure 1. Schematic (not to scale) of the experiment setup. The vehicle was driven at constant load 80 km/h for different periods of time during the feeding period on a chassis type dynamometer. Emissions were sampled at 6 LPM through a heated dilution and sampling system using two stage dilution systems, including porous tube dilutor (PTD) and ejector dilutor (ED), into ILMARI environmental chamber (Leskinen et al., 2015). The raw exhaust was sampled with the FTIR.**



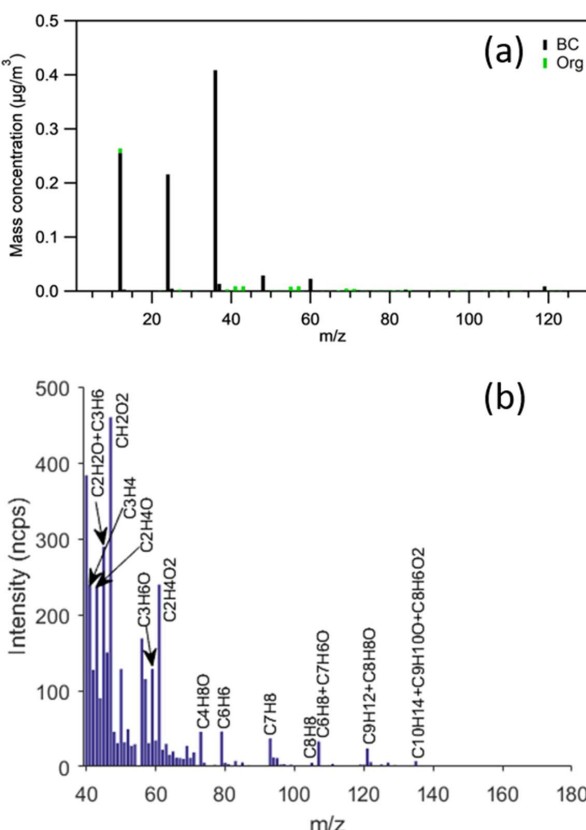

**Figure 2. The average particle-phase mass spectrum of pure vehicle exhaust measured inside the chamber right after the feeding period, before BL lamps were switched on (AMS data, panel a). The average gas-phase mass spectrum of all Pure vehicle and Mixed experiments after the exhaust feeding period (PTR-ToF-MS data, panel b). The instrument background measured through active carbon is subtracted from the gas-phase mass spectrum, and the water triplet signal from m/z 55 is removed for clarity. Gas-phase mass spectrum intensity is normalized against the primary ion ($H_3O^+$) signal intensity of $10^6$ cps.**



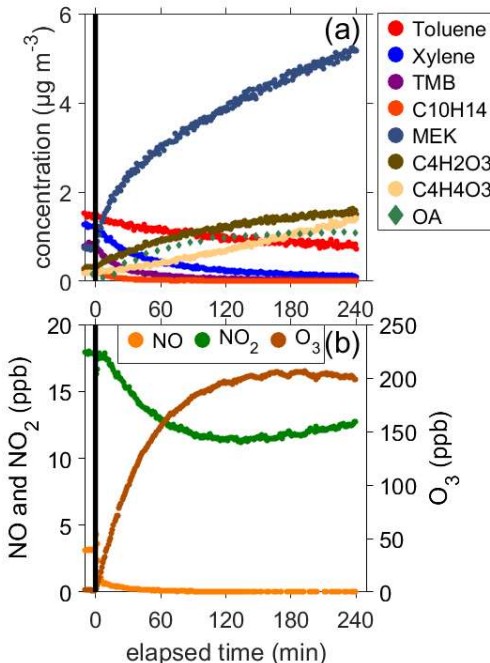

**Figure 3. Temporal evolution of GDI vehicle emitted VOCs, gas-phase oxidation products, and organic aerosol (panel a), and O₃ and nitrogen oxides (panel b) during Pure Vehicle1 experiment. At time zero the photochemistry experiment was started when BL lamps were switched on.**





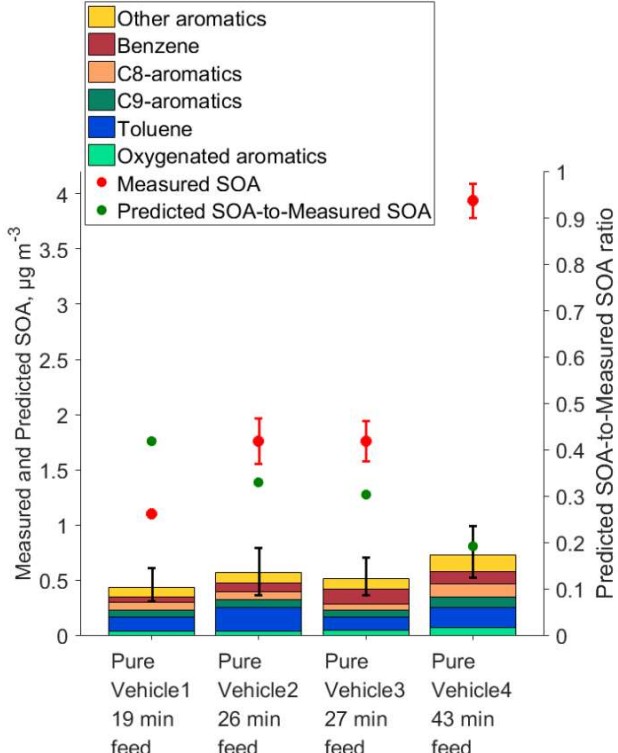

**Figure 4. Measured (red circles) and predicted (bars) SOA formed from the photochemistry of GDI vehicle exhaust. High NO$_x$ regime yields shown in Table S2 were applied to calculate the predicted SOA. Identified SOA precursors were divided into different groups to better illustrate the contribution of different types of VOCs to SOA formation. Error bars for predicted SOA represent lower and upper bound estimates and these were calculated by using SOA yields from different studies reported under similar conditions, representative for this study (see Table S2). Green circles illustrate Predicted SOA-to-Measured SOA ratio.**





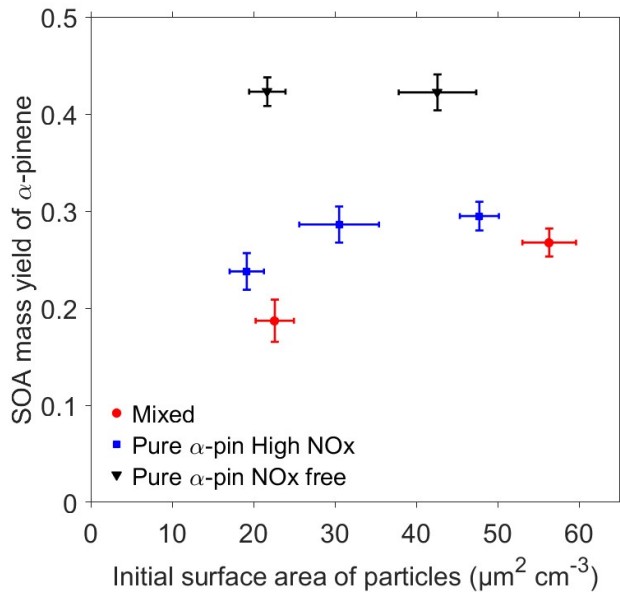

**Figure 5. α-Pinene SOA mass yield as a function of initial surface area of particles from all experiments. Error bars for the x-axis are estimated from the scatter in measurements of surface area of particles monitored by SMPS, and for the y-axis from uncertainties in SP-AMS and PTR-ToF-MS measurements using square root of sum of squares of relative standard deviations.**





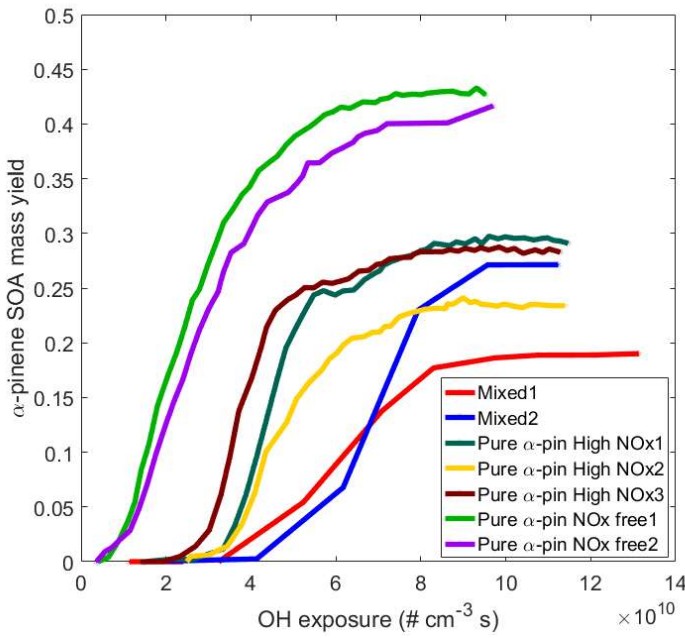

**Figure 6. α-Pinene SOA mass yield as a function of OH exposure from all experiments. To calculate α-pinene SOA mass yield for Mixed experiments, an assumption was made that ASOA contributed the same fraction from total SOA throughout the experiment.**



**Table 1**. Instrumentation used in this study.

| Instrument | Manufacturer, Model, Country | Measurement[a) |
|---|---|---|
| Soot particle- aerosol mass spectrometer (SP-AMS) | Aerodyne Research Inc., USA | Real-time measurements of particle composition. |
| Proton-transfer-reaction-time-of- flight-mass spectrometer (PTR-ToF-MS) | Ionicon Analytik, PTR-TOF 8000, Austria | Real-time measurements of gas-phase compounds. |
| acetate-chemical ionization time-of-flight-mass spectrometer (acetate-ToF-CIMS) | Aerodyne Research Inc., USA | Real-time measurements of acidic gas-phase compounds. |
| Trace-level chemiluminescence $NO-NO_2-NO_x$ analyzer | Thermo 42i-TL, USA | Continuous monitoring of $NO$, $NO_2$, and $NO_x$ concentrations. |
| UV-photometric ozone analyzer | Thermo 49i, USA | Continuous monitoring of $O_3$ concentration. |
| Trace-level pulsed fluorescence $SO_2$ analyzer | Thermo 43i, USA | Continuous monitoring of $SO_2$ concentration. |
| Fourier Transform Infrared Spectroscopy (FTIR) | Gasmet, Finland | Continuous measurements of $CO_2$, $CO$, and THC concentrations from raw, undiluted exhaust. |
| Scanning mobility particle sizer (SMPS) | TSI, 3081 DMA + C3775 CPC, USA | Particle size distributions. |
| Temperature and humidity probe | Vaisala HMP 60, Finland | Monitoring of relative humidity, and temperature. |
| Carbon dioxide sensor | Vaisala GMP 343, Finland | Measurement of concentration of $CO_2$. |

a) All measurements were collected inside the environmental chamber unless indicated otherwise.



**Table 2.** Experimental conditions of the experiments conducted in this study.

| Experiment | VOC-to-NOx (ppbC/ppb) | NO₂-to-NO | NO (ppb) | NO₂[a] (ppb) | SO₂ (ppb) | OH exposure (#/cm3 s) | Initial number conc. (#/cm3) | Initial surface area (µm2/cm3) | THC+OxyHC in raw exhaust (ppm) | Feeding time (min) | dilution factor |
|---|---|---|---|---|---|---|---|---|---|---|---|
| Pure Vehicle 1 | 8.6 | 5.6 | 3.2 | 17.8 | 2.3 | 2.08E+11 | 2.62E+03 | 6.31E+01 | 28.9 | 19 | 228 |
| Pure Vehicle 2 | 7.9 | 5.5 | 6.0 | 33.1 | 2.7 | 2.27E+11 | 3.42E+03 | 6.65E+01 | 28.4 | 26 | 199 |
| Pure Vehicle 3 | 5.9 | 3.4 | 14.1 | 48.8 | 1.9 | 2.64E+11 | 1.76E+03 | 3.71E+01 | 24.4 | 27 | 202 |
| Pure Vehicle 4 | 6.5 | 3.8 | 17.7 | 67.9 | 3.7 | 2.62E+11 | 3.94E+03 | 8.68E+01 | 24.5 | 43 | 203 |
| Mixed 1 | 6.3 | 3.2 | 16.7 | 53.5 | 2.8 | 2.51E+11 | 9.80E+02 | 2.26E+01 | 26.1 | 21 | 210 |
| Mixed 2 | 5.6 | 4.0 | 14.4 | 57.7 | 1.2 | 2.84E+11 | 2.29E+03 | 5.63E+01 | 28.2 | 22 | 220 |
| Pure α-pin high NOx 1 | 4.6 | 3.1 | 15.7 | 48.2 | - | 1.21E+11 | 1.64E+03 | 4.77E+01 | - | - | - |
| Pure α-pin high NOx 2 | 4.8 | 3.3 | 14.9 | 49.9 | - | 1.16E+11 | 7.85E+02 | 1.92E+01 | - | - | - |
| Pure α-pin high NOx 3 | 5.7 | 2.9 | 15.2 | 44.5 | - | 1.21E+11 | 1.22E+03 | 3.05E+01 | - | - | - |
| Pure α-pin NOx free 1 | - | - | - | - | - | 1.23E+11 | 1.57E+03 | 4.26E+01 | - | - | - |
| Pure α-pin NOx free 2 | - | - | - | - | - | 1.26E+11 | 7.85E+02 | 2.17E+01 | - | - | - |

a) In Pure Vehicle experiments NO₂ present inside the chamber before photochemistry period was produced from NO, i.e. no additional NOx was added in Pure Vehicle experiments.

5  **Table 3.** Vehicle derived SOA formed in Mixed experiments estimated with different approaches described in Sect. 3.3. RSD=relative standard deviation

| Experiment | ASOA Method 1 (µg m⁻³) | ASOA Method 2 (µg m⁻³) | ASOA Method 3 (µg m⁻³) | Average (µg m⁻³) | RSD (%) | ASOA fraction of total SOA (%) |
|---|---|---|---|---|---|---|
| Mixed1 | 0.97 | 1.03 | 1.24 | 1.08 | 13 | 20 |
| Mixed2 | 1.70 | 1.62 | 1.96 | 1.76 | 10 | 18 |