# Peer review of "Potential dual effect of anthropogenic emissions on the formation of biogenic SOA"

_Atmospheric Chemistry and Physics, 2019_

## Referee Comment (RC1) · Anonymous Referee #1 · 17 Jul 2019

The authors present a description of experiments in which a gasoline vehicle was run at constant speed and the exhaust was led into a chamber to perform photo-oxidation experiments. In addition, reference experiments, in which a biogenic model compound (alpha-pinene) was led into the same chamber together with NOx and without NOx, are presented. Finally, also experiments in which alpha-pinene and gasoline vehicle exhaust are led into the chamber together are presented.

Based on the experiments, the authors present results on the secondary aerosol formation process from both the vehicle exhaust and the biogenic aerosol precursors. Firstly, they present characterisations of the gasoline exhaust-produced SOA, and show that only a minority of the produced aerosol can be explained by precursors identified in the measurements, while the majority is produced from unidentified sources. Secondly, the

authors show data with the purpose of showing two different mechanisms causing the secondary organic aerosol production from biogenic precursors to be lower when gasoline car exhaust is present. The first mechanism is the effect of NOx on the emissions, which is expected as it has been seen earlier. The second effect is more novel, as the authors state that the anthropogenic VOCs change the reaction pathways, leading to lower yields.

I think that the experiments are very interesting and have been performed carefully, and the results are certainly of interest to aerosol scientists. However, I think that in the current form, the manuscript somewhat overestimates the magnitude of the second SOA suppression effect, and it is also lacking a more comprehensive discussion of possible other explanations that might cause the observed phenomena. There are several questions in relation to the evidence of the anthropogenic VOC effect that I think should be addressed before publication in ACP.

* from table 2 it seems that the NOx values were clearly (70.2 vs 64.8 for the low-surface area and 72.1 vs 63.9 for more surface area) higher in the mixed cases than in the alpha-pinene cases, by a factor of 10% in the higher surface area case. In the latter case, the difference seems to be of the same order than the difference in the yields between the two cases. Although it is not certain whether the suppression of SOA formation caused by NOx is directly dependent on the NOx concentration, I think it should be explored whether the suppression could be caused by this difference.

* The difference between the high-NOX alpha-pinene and the mixed case seems to be larger in the case of less initial particle surface. From the paper it was not directly evident whether there was formation of particles in the experiments (in addition to the existing seed particles). Is this the case? If yes, was there a difference between the different experiments in the number of particles formed? As this might change the dynamics of the gas-to-particle transfer, it would seem that the most relevant normalisation for the surface area (e.g. in Figure 5) would be the surface area at the time when the particles are being formed, i.e. during the time of the steepest increase in the yield

ACPD

in Figure 6. Would it be possible to produce such a figure, and is the result still similar (or even more clear) than when using the initial surface area?

* In figure 6, the second mixed experiment (which has a higher surface area) starts off slower but then reaches a higher yield than the other mixed experiment and even higher than alpha-pinene experiments (although the latter has a higher surface area). Is there an explanation for this anomalous behaviour (the other lines do not cross each other)?

* The different delay for the mixed experiments when compared to the alpha-pinene experiments seems a key issue to me. The authors state that wall losses of SOA-forming vapors are an issue that influences the SOA yield in the chamber. I would also think that some fraction of the injected alpha-pinene is lost to dilution in the chamber. Would it be possible to make an estimate of the magnitude of such loss processes of the precursors, and estimate if these could cause the differences in the yields?

* The authors call the new effect the anthropogenic VOC effect. There are also other compounds than VOCs in vehicle exhaust. Could it be possible that e.g. sulphur compounds or other such constituents could be the cause of the suppression?

* I am not convinced by the argument related to figure S5. For the mixed datapoint 5, I think it is evident that the mixed case produces less SOA than the combinations of vehicle and alpha-pinene measurements. However, for the data point 6, the authors choose a single comparison (points 6 and 21); however, one could as easily choose datapoints 20 and 6 and argue that actually the mixed experiment produced more SOA. I think that the purpose is to show that the mixed experiments lie on the lower edge of a 'line', but especially for the second experiment this does not fulfil the purpose, and I would either remove this figure or make i much more clear how it adds more evidence.

Based on these above points, I think that the claim of a having found a dual effect of the anthropogenic emissions should be argued more convincingly. Especially the effect of the losses and potential sources for error, and also the effect of the different NOx levels

should be discussed. The presence of a compound in the mixed case that is not seen in the other cases is nice evidence of a changed chemistry, but the conclusions that can be drawn from the data points are still quite speculative and there is quite some doubt on the magnitude of the effect.

In total, the manuscript should possibly use a more careful wording, and maybe change the title to "Potential dual effect of..." Also, sentences that state that alpha-pinene oxidation pathways have changed in the presence of vehicle exhaust (e.g. in the abstract) should be reworded so that it is clear that this is speculation.

The following points should also be clarified:

p 13, line 7: "However, these SOA precursors were not detected by the PTR-ToF-MS, most likely due to sampling line and instrumental losses (Pagonis et al., 2017). "

It is my understanding that the PTR-MS is not really suitable to compounds that have lower volatilities in general, also partly due to the ionisation mechanism (see eg. Riva et al., 2019) This is not really reflected in this sentence; if it was mainly a loss issue, there could still be a signal that would in general be proportional to the concentration. This could be clarified.

p13, line 18; "Our results imply that the contribution of IVOCs and SVOCs to formed SOA is driving time dependent, at least when the modern gasoline vehicle is driven at constant load."

To my understanding, there might also be other factors explaining the difference in a chamber experiment situation. SVOCs and IVOCs might be lost on the chamber walls at a different rate than VOCs; this is already implied in the section that my previous comment refers to. I think that the potential effects of wall losses should be discussed and maybe some reservation could be made in the text. Also, could a similar figure as Fig. 6 (with the amount of SOA as a function of the OH exposure) be shown to see if there is a difference in the 'onset' time of SOA formation?

Reference: Riva et al., Atmos. Meas. Tech., 12, 2403–2421, 2019 https://doi.org/10.5194/amt-12-2403-2019

---

## Referee Comment (RC2) · Anonymous Referee #2 · 22 Jul 2019

In this work, the authors characterise emissions from a modern GDI vehicle running at a constant load and investigate their SOA formation potential and their effect on SOA formation from a-pinene (used here as a model for biogenic emissions). The study concluded that the precursors measured by PTR-ToF-MS could only account for a fraction of the total SOA formed and concluded that lower volatility VOCs, not measured in this work, was likely to be a major contributor to SOA formation. It also reported a suppression of the a-pinene SOA mass yield when mixed with the anthropogenic emissions from the GDI engine and attempted to explain the main effects causing this suppression as "NOx" and "anthropogenic" effects. The NOx effect is clearly demonstrated through the set of experiments conducted and presented and it is consistent with what is known and reported in the literature. However, the "anthropogenic effect" reported by the authors is not sufficiently supported by the data presented in the current manuscript. The evidence for this interpretation is weak and not convincing given the limited number of mixed experiments and the lack of consistent results in Figure 4. The further reduction in a-pinene SOA mass yield is only shown for one of the two mixed experiments. The effect is not observed for the second mixed experiment compared to the pure a-pinene high NOx results. Although the authors attempted to attribute this to the effect of the initial surface area of particles, the surface area influence does not appear to be evident in the pure a-pinene NOx free experiments and in two of the three pure a-pinene high NOx experiments. Additionally, the effect of competition for oxidant in the mixed experiments has not been discussed as a potential reason for the changes observed in these work. As the "anthropogenic effect" is presented as one of the key "dual" effects of mixing anthropogenic and biogenic precursors, the current manuscript should not be accepted for publication in ACP in its current state and major revisions should be made to re-interpret the main findings before it could be considered for publication.

**Specific comments:**

Page 2, 13: most of our knowledge, to date, on the detrimental effects of aerosols on human health is related to PM2.5 or PM10 based on epidemiological studies. The effect of individual chemical components or classes is very plausible and often speculated on but it has not been yet fully established. Mixing the effect of SOA with the effect of total aerosols is a common practice but should be corrected until further evidence is established.

Page 2, 20: comment on the fuel sulfur content used in this study. This is also mentioned again on page 10 and should be qualified there too.

Page 5, 28: Specify the light characteristics during this work. The total actinic flux and photolysis rates of NO2 and O1D should be stated.

Page 6, line 32: elaborate on what is meant by "atmospherically relevant VOC/NOx ratios" reported in this study, Are the numbers in Table 2 based on the amount of VOCs

measured by the PTR for a specific number of compounds?. As discussed in the manuscript, these are only a subset of the total VOC present. This should be clarified in the manuscript.

Page 7, line 4: justify the choice of adding 5ppb of a-pinene in relation to the amount of AVOCs available from the emissions in terms of their potential to compete for the oxidants available. A quick calculation based on numbers in Table 2, indicate that the total VOC available in the experiments ranged from around 180 to 560ppb.

Page 7, 25: The Hao et al., method used for particle wall loss corrections assumed that particle wall loss rate constant is independent of size. The effect of size-resolved loss correction on total mass and SOA yield should be evaluated and reported.

Page 7, section 3.2.2: This approach adopted in this section is very simplistic and assumes that the SOA formation is an additive process and it ignores any potential non-linear interactions such as competition for oxidant or effect on product yields as recently demonstrated in McFiggans et al., 2019. Although some of these effects are later referred to in the text, stronger emphasis should be made earlier in the paragraph on these potential effects and the purpose of this analysis should be stated more clearly.

Page 11: section 3.1 appears to attempt to comment on the composition of the gas and condensed phase of the GDI exhaust. However, the supporting figures do not really support the overall message of the paragraph. The section needs more discussion including wider engagement with the relevant literature. The section lacks clear quantitative observations. For example, the statement made on line 15 of page 1 is not really supported by the data in the Figure or in table 2. I suggest that initial values for BC and organic matter should be included in Table 2.

Page 11, 17: Table 2 does not reflect the NOx emissions form the engine as it appears to report the values after the addition of ozone and NO2 top up as stated in the text. Therefore, the statement about "significant" amounts of NOx from the GDI engine cannot be made based on this data.

Page 11, 19: quantify what you mean by atmospherically relevant NO2/NO and VOC/Nox ratio and link it to a specific type of environment.

Page 11, 29: elaborate more on what you mean conditions were similar to Barcelona or Rome!!. This statement appears "out-of-the-blue" and is not supported.

Page 12, 7: The statements made about SOA and POA in pure vehicle exhaust experiments need to be supported by data. These should be presented in a table or in a summary plot.

Page 12, 12: the classification of "high NOx" experiments should be placed within the context of what has been discussed in the literature of high/low NOx conditions. This should not be based only on the absolute amount of NOx as it should take into consideration the VOC/NOx ratio. As it stands, the definition used in the manuscript can be confusing or misleading when thought about in the wider context of the literature on high/low NOx experiments.

Page 12, 14: This statement is not really supported by the data shown in the figure 3. The formation of O3 and titration of NO appeared to happen almost immediately after lights on and SOA build up didn't take very long at all to begin.

Page 18, 11: what is the source of ammonium nitrate in these experiments?

Page 18, 14: Ratio between 1.5 up to 3 for NO+/NO2+ is quite typical of ammonium nitrate depending on which AMS instrument is being used. The ratio for organic nitrate is typically larger than 5. The presented data show a very limited evidence for the formation of organic nitrate.

**Editorial comments:**

Page 3, 4: change "that" to "the"

Page 3, 9: change "affects greatly" to "greatly affects"

Page 3, 19: delete "an"

Page 3, 21: change "in" to "on"

Page 7, line 4: Express the 3 ul of butanol in ppb similar to what has been done for a-pinene

Page 8, 19: add model and manufacturer for the PTR

Page 9, 13: specify the instrument's resolution

Page 9, 30: delete "in the campaign"

Page 9, 30: change "switched" to "switching"

Figures 2 and 3 should be improved. They are currently of poor visual quality.
* * *

---

## Author Comment (AC1) · 16 Sep 2019

The authors present a description of experiments in which a gasoline vehicle was run at constant speed and the exhaust was led into a chamber to perform photo-oxidation experiments. In addition, reference experiments, in which a biogenic model compound (alpha-pinene) was led into the same chamber together with NOx and without NOx, are presented. Finally, also experiments in which alpha-pinene and gasoline vehicle exhaust are led into the chamber together are presented. Based on the experiments, the authors present results on the secondary aerosol formation process from both the vehicle exhaust and the biogenic aerosol precursors. Firstly, they present characterisations of the gasoline exhaust-produced SOA, and show that only a minority of the produced aerosol can be explained by precursors identified in the measurements, while the majority is produced from unidentified sources. Secondly, the authors show data with the purpose of showing two different mechanisms causing the secondary organic aerosol production from biogenic precursors to be lower when gasoline car exhaust is present. The first mechanism is the effect of NOx on the emissions, which is expected as it has been seen earlier. The second effect is more novel, as the authors state that the anthropogenic VOCs change the reaction pathways, leading to lower yields. I think that the experiments are very interesting and have been performed carefully, and the results are certainly of interest to aerosol scientists. However, I think that in the current form, the manuscript somewhat overestimates the magnitude of the second SOA suppression effect, and it is also lacking a more comprehensive discussion of possible other explanations that might cause the observed phenomena. There are several questions in relation to the evidence of the anthropogenic VOC effect that I think should be addressed before publication in ACP.

We thank the reviewer for his/ her comprehensive and helpful comments. Below we answer point by point the questions and criticism raised by the reviewer.

* from table 2 it seems that the NOx values were clearly (70.2 vs 64.8 for the low-surface area and 72.1 vs 63.9 for more surface area) higher in the mixed cases than in the alpha-pinene cases, by a factor of 10% in the higher surface area case. In the latter case, the difference seems to be of the same order than the difference in the yields between the two cases. Although it is not certain whether the suppression of SOA formation caused by NOx is directly dependent on the NOx concentration, I think it should be explored whether the suppression could be caused by this difference.

Indeed, there was a difference in NOx concentration between different experiments, as our main target was to try to adjust VOC-to-NOx concentrations as well as possible. As the referee points out, there's no clear evidence showing that NOx concentration itself would affect the yield if the VOC-to-NOx ratios are equal. According to earlier studied comparable VOC-to-NOx ratios between the experiments is a critical factor in SOA formation (see e.g. Presto et al., 2005, Hoyle et al., 2011). We would like to point out that actually in the Pure α-pinene- High NOx experiments we had slightly lower VOC-to-NO$_x$ ratios than in Mixed experiments as shown in Table 2 (Pure apinene experiments: 4.6 - 4.8 - 7.5; Mixed 5.6 – 6.3). This should lead to slightly decreased α-pinene SOA mass yields in Pure α-pinene- High NOx experiments compared to Mixed experiments (Presto et al., 2005). The difference in VOC-to-NOx ratios are actually larger, than in NOx concentrations, hence we believe that the VOC-to-NOx ratio difference should dominate over NOx concentration difference and this should result in lower yield in Pure α-pinene- High NOx compared to Mixed experiments.

* The difference between the high-NOX alpha-pinene and the mixed case seems to be larger in the case of less initial particle surface. From the paper it was not directly evident whether there was formation of particles in the experiments (in addition to the existing seed particles). Is this the case? If yes, was there a difference between the different experiments in the number of particles formed? As this might change the dynamics of the gas-to-particle transfer, it would seem that the most relevant normalisation for the surface area (e.g. in Figure 5) would be the surface area at the time when the particles are being formed, i.e. during the time of the steepest increase in the yield in Figure 6. Would it be possible to produce such a figure, and is the result still similar (or even more clear) than when using the initial surface area?

There was a nucleation in the case of less initial particle surface. We did re- plot the Figure 5. using the surface area in the beginning of the SOA formation (see the Figure below). But as the steepness of the curves in Figure 6 are quite different between different experiments, it turned out to be challenging, to define the point that would physically correspond each other in different experiments. This resulted in large uncertainties of the initial surface area as can be seen in the figure below.

Figure also shows, that the similar trend can be seen in the replotted Figure as in the original Figure 5., so we decided to keep the original Fig, 5 in the manuscript.

[Figure]

* In figure 6, the second mixed experiment (which has a higher surface area) starts off slower but then reaches a higher yield than the other mixed experiment and even higher than alpha-pinene experiments (although the latter has a higher surface area). Is there an explanation for this anomalous behaviour (the other lines do not cross each other)?

Yes, this is interesting point, and we tried to find an answer to this question already when analyzing the data. Unfortunately, we don't have a clear explanation to this behavior.

* The different delay for the mixed experiments when compared to the alpha-pinene experiments seems a key issue to me. The authors state that wall losses of SOA-forming vapors are an issue that influences the SOA yield in the chamber. I would also think that some fraction of the injected alpha-pinene is lost to dilution in the chamber. Would it be possible to make an estimate of the magnitude of such loss processes of the precursors, and estimate if these could cause the differences in the yields?

The dilution effect should not affect the results, since we measured the α-pinene concentration inside the chamber by the PTR-MS. Hence the potential differences in the dilution reflects to measured α-pinene concentrations. α-pinene vapor pressure is so high (on the order of $10^4$ µg/m3), that wall losses are negligible as shown e.g. in Kokkola et al., 2014. We would like to highlight that the chamber used in this study has larger volume-to-surface ratio than the one used in Kokkola et al. study, hence the wall losses are even smaller than those modelled and measured in Kokkola et al.

* The authors call the new effect the anthropogenic VOC effect. There are also other compounds than VOCs in vehicle exhaust. Could it be possible that e.g. Sulphur compounds or other such constituents could be the cause of the suppression?*

We did not detect Sulphur compounds in our experiments most likely due to the low concentration of $SO_2$ introduced into the chamber by the vehicle exhaust (see Table 2). However, we cannot rule out the possibility of Sulphur compounds contributing to the additional suppression of α-pinene SOA yields, but our data indicates that the main reason for this suppression (in addition to NOx) would have been anthropogenic VOCs. We are well aware that there might be other possible explanations for this additional suppression, and we are pointing out in the manuscript (for example page 18, line 32) that we cannot rule out other possible suppression mechanisms, given the limitations of the instrumentation used (and data obtained) in this study. We have now added a mention of $SO_2$ to the text. (page 20).

I am not convinced by the argument related to figure S5. For the mixed datapoint 5,I think it is evident that the mixed case produces less SOA than the combinations of vehicle and alpha-pinene measurements. However, for the data point 6, the authors choose a single comparison (points 6 and 21); however, one could as easily choose datapoints 20 and 6 and argue that actually the mixed experiment produced more SOA.I think that the purpose is to show that the mixed experiments lie on the lower edge of a 'line', but especially for the second experiment this does not fulfil the purpose, and I would either remove this figure or make i much more clear how it adds more evidence.

Indeed, our point was to show that the mixed experiments lie in the lower edge of the line. We do agree with the reviewer that the figure is more confusing than informative. Hence we have removed the Figure S5.

Based on these above points, I think that the claim of a having found a dual effect of the anthropogenic emissions should be argued more convincingly. Especially the effect of the losses and potential sources for error, and also the effect of the different NOx levels should be discussed. The presence of a compound in the mixed case that is not seen in the other cases is nice evidence of a changed chemistry, but the conclusions that can be drawn from the data points are still quite speculative and there is quite some doubt on the magnitude of the effect.

In total, the manuscript should possibly use a more careful wording, and maybe change the title to "Potential dual effect of..." Also, sentences that state that alpha-pinene oxidation pathways have changed in the presence of vehicle exhaust (e.g. in the abstract) should be reworded so that it is clear that this is speculation.

Taking into account our dataset and the underlying experimental uncertainties, we do agree that our presented results are only indicative, and hence we have reworded the parts of the manuscript that discuss about the dual effect of the vehicle exhaust on α-pinene SOA mass yields. We hope that these modifications make it more clear to the reader that the conclusions involve a certain degree of speculation and that more experiments are required in future studies to verify the suggested "anthropogenic" effect. We have also modified the title accordingly.

The following points should also be clarified:
p 13, line 7: "However, these SOA precursors were not detected by the PTR-ToF-MS, most likely due to sampling line and instrumental losses (Pagonis et al., 2017). "It is my understanding that the PTR-MS is not really suitable to compounds that have lower volatilities in general, also partly due to the ionisation mechanism (see eg. Riva et al., 2019) This is not really reflected in this sentence; if it was mainly a loss issue, there could still be a signal that would in general be proportional to the concentration. This could be clarified.

The reviewer is right that PTR-ToF-MS is not the most optimal instrument to measure lower volatility compounds. However, the main reason for this does appear to be instrumental and line losses. As long as lower volatility compounds have higher proton affinity than water the ionization would take place in the ionization chamber (if it were not for those proposed losses). The line and instrumental loss issue with the PTR-ToF-MS was also pointed out by Riva et al. (2019) "*As discussed earlier, the inlet of the PTR-TOF is not well enough designed to sample OVOCs with low volatility, which explained the lack of correlations for larger and more oxidized products between the PTR-TOF and the nitrate CI-APi-TOF.*" In the same study Riva et al. were in fact able to detect a large range of OVOCs (relatively low volatile compounds), by using Vocus-PTR that uses proton-transfer ionization mechanism but employs a different type of inlet design, which minimizes the inlet line and instrumental losses. In addition, Breitenlechner et al. (2017) state in their publication that "Existing PTR-TOF instruments are known do detect VOC and could in principle also detect highly oxidized organic compounds such as LVOC and ELVOC but PTR-TOF-MS inlets were not optimized to avoid wall losses of such low volatility compounds. In addition PTR-TOF-MS is not sensitive enough to quantify second order and even higher order oxidation products at atmospherically relevant concentrations." (2017, Analytical chemistry) To solve this problem, Breitenlechner et al. developed the PTR3 instrument that (like the Vocus-PTR) is capable to measure low volatile oxidation products using proton-transfer-reaction chemical ionization method. Therefore, it is likely that instrumental and

line losses were the main reasons why we did not detect low volatility compounds with the PTR-ToF-MS used in this study. Anyhow, we have clarified the sentence so that we are also mentioning now the fact that the detection efficiency depends naturally on proton affinity of the compounds, hence only compounds having proton affinity higher than water can be detected.

p13, line 18; "Our results imply that the contribution of IVOCs and SVOCs to formed SOA is driving time dependent, at least when the modern gasoline vehicle is driven at constant load. "To my understanding, there might also be other factors explaining the difference in a chamber experiment situation. SVOCs and IVOCs might be lost on the chamber walls at a different rate than VOCs; this is already implied in the section that my previous comment refers to. I think that the potential effects of wall losses should be discussed and maybe some reservation could be made in the text.

We agree with the reviewer that the loss rate of vapors may play a role here. We have now mentioned this in the text.

Also, could a similar figure as Fig. 6 (with the amount of SOA as a function of the OH exposure) be shown to see if there is a difference in the 'onset' time of SOA formation?

Below we are presenting formed amount of SOA from a-pinene photo-oxidation as a function of OH exposure as reviewer suggested. The figure shows quite a similar behavior compared to Figure 6 of the manuscript (Fig 6 showed also below for comparison purposes). For example, the similar delay in start of SOA formation is observed.

[Figure]

**α-Pinene SOA as a function of OH exposure from all experiments**

[Figure]

**α-Pinene SOA mass yield as a function of OH exposure from all experiments (manuscript Figure 6).**

**References:**

Breitenlechner, M., Fischer, L., Hainer M., Heinritzi M., Curtius J., and Hansel A.: The PTR3: A novel instrument for studying the lifecycle of reactive organic carbon in the atmosohere. Analytical chemistry, 89, 5824-5831, 2017. DOI: 10.1021/acs.analchem.6b05110.

Hoyle, C. R., Boy, M., Donahue, N. M., Fry, J. L., Glasius, M., Guenther, A., Hallar, A. G., Hartz, K. H., Petters, M. D., Petaja, T., Rosenoern, T., and Sullivan, A. P.: A review of the anthropogenic influence on biogenic secondary organic aerosol, Atmos Chem Phys, 11, 321-343, 10.5194/acp-11-321-2011, 2011.

Kokkola, H., Yli-Pirilä, P., Vesterinen, M., Korhonen, H., Keskinen, H., Romakkaniemi, S., Hao, L., Joutsensaari, J., Worsnop, D.R., Virtanen, A., and Lehtinen, K.E.J. The role of low volatile organics on secondary organic aerosol formation. Atmos. Chem. Phys., 14, 1689-1700, 2014.

Presto, A. A., Hartz, K. E. H., and Donahue, N. M.: Secondary organic aerosol production from terpene ozonolysis. 2. Effect of NOx concentration, Environ Sci Technol, 39, 7046-7054, 10.1021/es050400s, 2005.

Riva M., Rantala P., Krechmer J. E., Peräkylä O., Zhang Y., Heikkinen L., Garmash O., Yan C., Kulmala M., Worsnop D., Ehn M.: Evaluating the performance of five different chemical ionization techniques for detecting gaseous oxygenated organic species. Atmos. Meas. Tech, 12, 2403-2421, 2019.

---

## Author Comment (AC2) · 16 Sep 2019

In this work, the authors characterise emissions from a modern GDI vehicle running at a constant load and investigate their SOA formation potential and their effect on SOA formation from a-pinene (used here as a model for biogenic emissions). The study concluded that the precursors measured by PTR-ToF-MS could only account for a fraction of the total SOA formed and concluded that lower volatility VOCs, not measured in this work, was likely to be a major contributor to SOA formation. It also reported a suppression of the a-pinene SOA mass yield when mixed with the anthropogenic emissions from the GDI engine and attempted to explain the main effects causing this suppression as "NOx" and "anthropogenic" effects. The NOx effect is clearly demonstrated through the set of experiments conducted and presented and it is consistent with what is known and reported in the literature. However, the "anthropogenic effect" reported by the authors is not sufficiently supported by the data presented in the current manuscript. The evidence for this interpretation is weak and not convincing given the limited number of mixed experiments and the lack of consistent results in Fig. 5. The further reduction in a-pinene SOA mass yield is only shown for one of the two mixed experiments. The effect is not observed for the second mixed experiment compared to the pure a-pinene high NOx results. Although the authors attempted to attribute this to the effect of the initial surface area of particles, the surface area influence does not appear to be evident in the pure a-pinene NOx free experiments and in two of the three pure a-pinene high NOx experiments.

The effect of seed surface concentration on SOA yield has been thoroughly discussed and shown in previous studies e.g. in Zhang et al., where this was demonstrated both by experimental data and theoretical approach. Also in our previous paper (Kari et al., 2017) we have shown by measurements a clear influence of seed surface area on a-pinene SOA yield in the same chamber used in this study. Two out of the three data sets in Figure 5., shows an increasing trend in SOA yield with increasing seed surface area. The clarity of this effect is somewhat blurred due to variations in experimental conditions (which are contributing to our the fairly large error bars). E.g. in "pure a-pin high NOx" (blue points in Fig. 5) experiments there was some variability in VOC-to-NOx ratios resulting in less clear trend. The reason why "Pure a-pin NOx-free" (black points in Fig 5) doesn't show an increasing trend in yield with increasing aerosol surface area is unclear, but we refer to the general experimental uncertainties related to the measurements. Taking into account both our dataset and the underlying experimental uncertainties, we do agree that our presented results are only indicative, and hence we have reworded the parts of the manuscript that discuss about the dual effect of the vehicle exhaust on α-pinene SOA mass yields. We hope that these modifications make it more clear to the reader that the concerned conclusions involve a certain degree of speculation and that more experiments are required in future studies to verify the suggested "anthropogenic" effect. We have also modified the title accordingly.

Additionally, the effect of competition for oxidant in the mixed experiments has not been discussed as a potential reason for the changes observed in these work.

The oxidant levels used in this study were high enough to ensure that we were not oxidant limited in the experiments.

As the "anthropogenic effect" is presented as one of the key "dual" effects of mixing anthropogenic and biogenic precursors, the current manuscript should not be accepted for publication in ACP in its current state and major revisions should be made to re-interpret the main findings before it could be considered for publication.

As indicated above, we have modified the title and rephrased the manuscript based on the criticism.

Specific comments:
Page 2, 13: most of our knowledge, to date, on the detrimental effects of aerosols on human health is related to PM2.5 or PM10 based on epidemiological studies. The effect of individual chemical components or classes is very plausible and often speculated on but it has not been yet fully established. Mixing the effect of SOA with the effect of total aerosols is a common practice but should be corrected until further evidence is established.

Thank you for pointing this out. We have corrected this accordingly.

Page 2, 20: comment on the fuel sulfur content used in this study. This is also mentioned again on page 10 and should be qualified there too.

We have reported the concentrations of $SO_2$ in each experiment in Table 2 of the original manuscript (now moved to new Table 3 in the revised manuscript).

Page 5, 28: Specify the light characteristics during this work. The total actinic flux and photolysis rates of NO2 and O1D should be stated.

The light characteristics were as follows: actinic flux was $9.5963\cdot10^{15}$ photons $cm^{-2}$ $s^{-1}$; NO2 photolysis rate was 0.0035 $s^{-1}$; O1D photolysis rate was $2.2130\cdot10^{-5}$ $s^{-1}$. We have added the information also to the revised manuscript.

Page 6, line 32: elaborate on what is meant by "atmospherically relevant VOC/NOx ratios" reported in this study, Are the numbers in Table 2 based on the amount of VOCs measured by the PTR for a specific number of compounds?. As discussed in the manuscript, these are only a subset of the total VOC present. This should be clarified in the manuscript.

Atmospherically relevant VOC-to-NOx ratio means the ratio observed in the real atmosphere that enables all radical branching channels to occur in smog chamber experiments similarly with the real atmosphere under certain conditions. The range for this ratio was taken from the review of Hoyle et al. (2011, ACP). VOCs were measured by FID to determine VOC-to-NOx ratio prior to photochemistry period. We have specified this in Table 1.

Page 7, line 4: justify the choice of adding 5ppb of a-pinene in relation to the amount of AVOCs available from the emissions in terms of their potential to compete for the oxidants available. A quick calculation based on numbers in Table 2, indicate that the total VOC available in the experiments ranged from around 180 to 560ppb.

First 180 ppb to 560 ppb reviewer is referring to are actually ppb(C) values measured by FID (as shown in Table 2). Hence, these values are not directly related to VOC concentrations (for example 5 ppb of α-pinene corresponds to 50 ppb(C) as α-pinene contains 10 carbon atoms). 5 ppb of α-pinene was chosen based on the first pure vehicle experiments. We found out that the GDI vehicle emission introduced in to the chamber was comprised of approximately 2 ppb of PTR-ToF-MS detectable aromatics (the main SOA precursors from VOCs detected in this study) when the feeding time of the exhaust was comparable between Pure vehicle and Mixed experiments. Obviously vehicle exhaust contained other VOCs as well that did not contribute to SOA formation but were present inside the chamber and underwent photo-oxidation. Hence, 5 ppb of α-pinene was chosen to have biogenic model compound in a comparable level with the aromatics (SOA forming anthropogenic VOCs).

Page 7, 25: The Hao et al., method used for particle wall loss corrections assumed that particle wall loss rate constant is independent of size. The effect of size-resolved loss correction on total mass and SOA yield should be evaluated and reported.

The Hao et al. method was used to correct the mass loss of particles to chamber walls, which is widely used method (e.g. Hao et al. 2011 ACP, Presto and Donahue 2006 Environ. Sci. Technol., Pathak et al. 2007 J. Geophys. Res-Atmos.). In this method, we assume that the aerosol mass wall loss is first order and that the loss rate constant is independent of particle size. The loss rate was estimated using the last 2 hours of data of the measurements to make sure that SOA formation had stopped inside the chamber and therefore would not interfere the wall loss correction. As shown in the figure below, the fit of ln(V) vs time is excellent ($R^2$=0.97), giving us confidence that the model assumptions of diameter-independent mass loss is valid for our data set. We have clarified this in the manuscript on page 7. Consequently, we think that we don't need to use an alternative method to correct particle wall losses suggested by the reviewer.

[Figure]

Fig. 1 Example how wall loss rate was determined. Example plot is taken from Mixed 2 experiment.

Page 7, section 3.2.2: This approach adopted in this section is very simplistic and assumes that the SOA formation is an additive process and it ignores any potential non-linear interactions such as competition for oxidant or effect on product yields as recently demonstrated in McFiggans et al., 2019. Although some of these effects are later referred to in the text, stronger emphasis should be made earlier in the paragraph on these potential effects and the purpose of this analysis should be stated more clearly.

We assume that the reviewer is talking about section 2.3.2. Reviewer is right that this approach is a simplified approach, but still the approach enables us to estimate within some uncertainty levels the formed amount of SOA from the identified SOA precursors. In addition, this kind of approach has been used in several previous publication (e.g. Du et al. 2018 ACP, Peng et al. 2017 ACP, Platt et al. 2013 ACP). Anthropogenic VOC mixtures and the potential non-linear interactions have not been systematically studied, and there is an obvious need for this kind of studies. We have now clarified the purpose of the analysis, and mentioned possible non-linear interaction, in the section 2.3.2.

Page 11: section 3.1 appears to attempt to comment on the composition of the gas and condensed phase of the GDI exhaust. However, the supporting figures do not really support the overall message of the paragraph. The section needs more discussion including wider engagement with the relevant literature. The section lacks clear quantitative observations. For example, the statement made on line 15 of page 1 is not really supported by the data in the Figure or in table 2. I suggest that initial values for BC and organic matter should be included in Table 2.

We have included the initial BC and organic matter values in new Table 3.

Page 11, 17: Table 2 does not reflect the NOx emissions form the engine as it appears to report the values after the addition of ozone and NO2 top up as stated in the text. Therefore, the statement about "significant" amounts of NOx from the GDI engine cannot be made based on this data.

We have added the NOx concentration emitted by GDI vehicle to new Table 3.

Page 11, 19: quantify what you mean by atmospherically relevant NO2/NO and VOC/Nox ratio and link it to a specific type of environment.

The ratios of 3-8 for VOC/NOx ratios and 3-6 for NO2/NO were applied in our experiments, referring to a typical suburban atmospheric environment (Seinfeld, 1991). We have added this to the revised manuscript.

- Seinfeld J.H., National Research Council. 1991. *Rethinking the Ozone Problem in Urban and Regional Air Pollution*. Washington, DC: The National Academies Press. https://doi.org/10.17226/1889.

[Figure]

Figure 8-13
VOC, NOx and ozone concentrations in the atmospheric boundary layer at four locations. VOC is shown as Propy-Equiv concentrations in units of ppbcarbon.

Page 11, 29: elaborate more on what you mean conditions were similar to Barcelona or Rome!!. This statement appears "out-of-the-blue" and is not supported.
The purpose of this statement was to highlight that our experimental conditions were representative to conditions found from the real atmosphere. References after this statement include information about the conditions in Rome and Barcelona that support the statement. Even though it's a bit unclear to us what the referee pointing with this comment, we clarified the sentence.

Page 12, 7: The statements made about SOA and POA in pure vehicle exhaust experiments need to be supported by data. These should be presented in a table or in a summary plot.

We have added this information to new Table 3 to the revised manuscript.

Page 12, 12: the classification of "high NOx" experiments should be placed within the context of what has been discussed in the literature of high/low NOx conditions. This should not be based only on the absolute amount of NOx as it should take into consideration the VOC/NOx ratio. As it stands, the definition used in the manuscript can be confusing or misleading when thought about in the wider context of the literature on high/low NOx experiments.
We agree with the reviewer that the term "high NOx" conditions should always take into consideration the VOC-to-NOx ratio. We have discussed about this already in the original version of the manuscript

that different groups have defined high or low NOx conditions differently and this affects predicted SOA reported in different studies (page 14). We have made the corrections to the manuscript as reviewer suggested (page 12 of the revised manuscript).

Page 12, 14: This statement is not really supported by the data shown in the figure 3.The formation of O3 and titration of NO appeared to happen almost immediately after lights on and SOA build up didn't take very long at all to begin.

It's not clear to us what the referee means by this comment, we state in the manuscript that "This additional reaction pathway causes the delay in the start of SOA formation as Figure 3 shows– SOA formation did not start until the most of NO had reacted." And this is clearly shown in Figure 3.

Page 18, 11: what is the source of ammonium nitrate in these experiments?

Ammonium nitrate was formed through the reaction of background ammonia ($NH_3$) in the chamber with nitric acid ($HNO_3$), which is produced in the photooxidation of NOx.

Page 18, 14: Ratio between 1.5 up to 3 for NO+/NO2+ is quite typical of ammonium nitrate depending on which AMS instrument is being used. The ratio for organic nitrate is typically larger than 5. The presented data show a very limited evidence for the formation of organic nitrate.

As stated in the manuscript, the NO+/NO2+ ratio of ammonium nitrate is 1.46±0.02. After UV lights were switched on, the ratio monotonically rise to 2 in the α-pinene NOx-free experiments and to 3 in the a-pinene high-NOx experiments. The rising values of NO+/NO2+ ratios in these experiments can't be interpreted by other reasons except for the formation of organic nitrates. We agree with referee that the ratio for organic nitrate is typically larger that 5. Hence, the values of 3 and 2 are measured as an outcome of mixed ammonium nitrate (lower NO+/NO2+ ratio) and organic nitrate (higher NO+/NO2+ ratio).

Editorial comments:
Corrected
Page 3, 4: change "that" to "the"Page 3, 9: change "affects greatly" to "greatly affects"
Page 3, 19: delete "an"
Page 3, 21: change "in" to "on"
Page 7, line 4: Express the 3 ul of butanol in ppb similar to what has been done for a-pinene
Page 8, 19: add model and manufacturer for the PTR
Page 9, 13: specify the instrument's resolution
Page 9, 30: delete "in the campaign"
Page 9, 30: change "switched" to "switching"
Figures 2 and 3 should be improved. They are currently of poor visual quality.